# Induction of the hepatic aryl hydrocarbon receptor by alcohol dysregulates autophagy and phospholipid metabolism via PPP2R2D

Yun Seok Kim [1,2,3], Bongsub Ko [1], Da Jung Kim [1,4], Jihoon Tak[3,5], Chang Yeob Han[3,6], Joo-Youn Cho [1,2], Won Kim [7] & Sang Geon Kim [5] ✉

Disturbed lipid metabolism precedes alcoholic liver injury. Whether and how AhR alters degradation of lipids, particularly phospho-/sphingo-lipids during alcohol exposure, was not explored. Here, we show that alcohol consumption in mice results in induction and activation of aryl hydrocarbon receptor (AhR) in the liver, and changes the hepatic phospho-/sphingo-lipids content. The levels of kynurenine, an endogenous AhR ligand, are elevated with increased hepatic tryptophan metabolic enzymes in alcohol-fed mice. Either alcohol or kynurenine treatment promotes AhR activation with autophagy dysregulation via AMPK. *Protein Phosphatase 2 Regulatory Subunit-Bdelta* (*Ppp2r2d*) is identified as a transcriptional target of AhR. Consequently, PPP2R2D-dependent AMPKα dephosphorylation causes autophagy inhibition and mitochondrial dysfunction. Hepatocyte-specific AhR ablation attenuates steatosis, which is associated with recovery of phospho-/sphingo-lipids content. Changes of AhR targets are corroborated using patient specimens. Overall, AhR induction by alcohol inhibits autophagy in hepatocytes through AMPKα, which is mediated by *Ppp2r2d* gene transactivation, revealing an AhR-dependent metabolism of phospho-/sphingo-lipids.

Alcoholic liver disease (ALD), as featured by an accumulation of neutral lipids and disruption of lipids metabolism preceding liver dysfunction and hepatitis, belongs to the most prevalent liver diseases worldwide[1,2]. The aryl hydrocarbon receptor (AhR), a receptor highly expressed in metabolically active tissues[3], contributes not only to the xenobiotic response but to the regulation of homeostatic processes[4]. Notably, kynurenine as a metabolite produced from tryptophan by indoleamine 2,3-dioxygenase (IDO) and/or tryptophan 2,3-dioxygenase (TDO2) serves as AhR ligand[5],

drawing the attention of AhR as a potential metabotropic receptor molecule.

Autophagy is a process necessary for the maintenance of intracellular lipid homeostasis. Thus, autophagy dysfunction is associated with liver diseases, obesity, and diabetes[6]. Nowadays, research focuses on an understanding of global changes in lipids profile. Since chronic alcohol abuse suppresses autophagy, resulting in lipid accumulation and damaged organelles[7], an increase in autophagy can attenuate ALD progression[8]. Despite the recent advances in this field, the pathogen-

[1]Department of Clinical Pharmacology and Therapeutics, Seoul National University College of Medicine, Seoul 03080, Korea. [2]Department of Biomedical Sciences, Seoul National University College of Medicine, Seoul 03080, Republic of Korea. [3]College of Pharmacy, Seoul National University, Seoul, Republic of Korea. [4]Metabolomics Core Facility, Department of Transdisciplinary Research and Collaboration, Biomedical Research Institute, Seoul National University Hospital, Seoul 03082, Korea. [5]College of Pharmacy and Integrated Research Institute for Drug Development, Dongguk University-Seoul, Goyang-si, Kyeonggi-do 10326, Republic of Korea. [6]School of Pharmacy and Institute of New Drug Development, Jeonbuk National University, 567 Baekje-daero, Deokjin-gu, Jeonju-si, Jeollabuk-do 54896, Korea. [7]Division of Gastroenterology and Hepatology, Department of Internal Medicine, Seoul National University College of Medicine, Seoul Metropolitan Government Seoul National University Boramae Medical Center, Seoul, Korea. ✉e-mail: sgkim@dongguk.edu

esis resulting from alcohol consumption and autophagy dysregulation remains to be further established, particularly in conjunction with biologically active lipids contributing to cell membrane physiology and signaling.

Previously, AhR has been shown to be related to the amelioration of liver lesions by regulating a prebiotic intestinal microbiota[9], and AhR activation may exert a protective response to counteract redox imbalance[10]. However, several methodologic differences might account for conflicting results between studies, such as the type of gene knockout, the length of alcohol exposure, and the concentration of alcohol either in vivo or in vitro, as each of these can reflect different stages in the progression of ALD. Moreover, none of the studies have elucidated the underlying mechanisms of AhR-mediated lipid metabolism via autophagy in animal or alcoholic patient liver tissues. In addition, current research has not established whether and how AhR alters lipid degradation and lipid profile changes, particularly phospho-/sphingo-lipids, in conjunction with autophagy dysregulation.

Here, we report that autophagy is impaired by AhR overinduction under alcohol consumption. Our results also demonstrate that the alcohol-mediated induction of tryptophan-metabolizing enzymes facilitates kynurenine production in hepatocytes, which promotes the overexpression and activation of AhR. Mechanistically, AhR activation inhibits AMP-activated protein kinase alpha (AMPKα) by inducing protein phosphatase 2 regulatory subunit B delta (*Ppp2r2d*), suppressing autophagy. Functionally, the inhibition of autophagy by AhR disturbs the catabolism of neutral lipids, which causes mitochondrial dysfunction. Moreover, AhR ablation prevented alcohol from disrupting the homeostatic metabolism of phospho-/sphingo-lipids. Finally, the targets identified in this study were determined using patient liver specimens.

## Results

### Alcohol induces AhR activation in hepatocytes

As a first step toward identifying the key regulators of ALD, pathways were analyzed using a public dataset. Notably, several AhR-related gene sets were upregulated among the top 19 KEGG pathways (Supplementary Fig. 1a, upper). In subsequent leading-edge analyses, overlaps were identified between AhR-related pathways and the associated genes (Supplementary Fig. 1a, lower). To understand alcohol-induced AhR activation in detail, RNA-sequencing (RNA-seq) analysis was conducted using livers from mice subjected to either control or the Lieber-DeCarli alcohol diet. Principal component analysis (PCA) of the gene expression levels revealed distinct segregation between the ethanol-exposed and control organisms (Fig. 1a, left). Moreover, a Pearson correlation matrix was generated using all 13,345 gene expression levels to compare the transcriptomes from each sample (Fig. 1a, right). The differentially expressed genes (DEGs) accounted for 5.7% of the entire transcriptomes. Among the 754 DEGs, 379 genes were downregulated, whereas 375 genes were upregulated (Fig. 1b). A gene expression heatmap further confirmed the top 119 DEGs (Supplementary Fig. 1b). In Gene Set Enrichment Analysis (GSEA) analysis, the "AhR pathway" was the most highly upregulated among the "WP pathways" (Fig. 1c). Among the top 15 KEGG pathways, AhR-related gene sets were enhanced (Fig. 1d, upper). Moreover, leading-edge analysis once again elucidated overlaps between AhR-related pathways and the related genes (Fig. 1d, lower).

Next, AhR expression was assessed in alcohol-exposed mice via qRT-PCR analysis. Alcohol exposure not only increased *Ahr* transcript levels but also activated AhR activity, as demonstrated by the induction of *Cyp1a1* (Fig. 1e). The bona fide overexpression and activation of AhR were corroborated by immunoblotting and immunohistochemistry (Fig. 1f, g). In addition, AhR-positive cells showed the expression of CYP1A1 in mouse liver (Supplementary Fig. 1c). Concordantly, AhR was highly enriched in the nuclear fraction of the alcohol-exposed mice, thus supporting the ability of alcohol to activate AhR (Fig. 1h).

Consistently, the AhR-activating effect of alcohol and the resulting *Cyp1a1* induction were verified using mPHs and AML-12 cells (Fig. 1i, j). In addition, ethanol treatment increased AhR levels in Kupffer cells, but resulted in the opposite effect in hepatic stellate cells (HSCs) (Supplementary Fig. 1d), as shown previously[11,12]. Our findings indicate that alcohol enhanced AhR expression and its activation in hepatocytes.

### Ablation of AhR in hepatocytes prevents alcohol from dysregulating autophagy flux

Next, we ablated the *Ahr* gene in hepatocytes using *Ahr^{fl/fl} Alb-Cre* mice (*Ahr* HKO). Given that the loxP sites flanking the second exon of AhR did not affect AhR expression or other phenotypes, wild-type (WT) littermates without the *Cre* gene were used as controls (Supplementary Fig. 2a). Consistent with previous reports[13], PCR analysis of representative tissues and mPHs to assess the occurrence of both *Ahr^{fl/fl}*-unexcised and *Ahr^{fl/fl}*-excised alleles confirmed hepatocyte-specific gene deletion (Supplementary Fig. 2b). Immunoblottings also confirmed AhR silencing in hepatocytes (Supplementary Fig. 2c). The *Ahr* HKO mice were born at the expected Mendelian frequency, fertile, and without any gross physical or behavioral abnormalities observable up to 12 months of age.

RNA-seq analyses were then conducted to identify the gene networks affected by *Ahr* HKO. AhR ablation caused major changes in gene expression compared with the control (Fig. 2a, left). Heatmap visualizations indicated that WT and *Ahr* HKO exhibited distinct gene expression patterns, of which three biological replicates were correlated. Statistical analyses were conducted to identify the total and top 100 DEGs between WT and *Ahr* HKO (Supplementary Fig. 2d, e). The DEGs accounted for 9.1% of the total transcriptomes. Among the 1217 DEGs, 703 genes were downregulated with 514 genes upregulated (Fig. 2a, right). Of the 514 genes upregulated by AhR deletion, 33 genes were related to autophagy pathways (Supplementary Fig. 2f). Moreover, several gene sets associated with autophagy were upregulated in *Ahr* HKO according to REACTOME pathway analysis (Fig. 2b).

Autophagy plays a crucial role in lipid metabolism[14,15], cellular and tissue homeostasis[6], and causes an increase of LC3B II with p62 degradation[16]. In our assays, alcohol treatment decreased LC3B II levels with an increase of p62, confirming reduced autophagy flux (Supplementary Fig. 2g, h), which is in line with previous reports[17,18]. Therefore, we hypothesized that alcohol dysregulates autophagy through AhR. WT mice fed with the Lieber-DeCarli alcohol diet displayed a decreased number of autophagic vesicles in hepatocytes, which was entirely reversed in *Ahr* HKO mice (Fig. 2c). This effect was confirmed by changes in LC3B and p62 levels (Fig. 2d, e). Similar outcomes were obtained in the ex vivo and in vitro experiments using mPHs isolated from mice fed with an alcohol diet (Fig. 2f) or either ethanol-treated mPHs or AML-12 cells (Fig. 2g), verifying that alcohol dysregulates autophagy through AhR. Unlike the negligible p62 protein expression in *Ahr* HKO mPHs (Fig. 2g), the levels of its mRNA was not changed by ablation of AhR (Supplementary Fig. 2i), supporting the notion that p62 may not be the direct target of AhR.

To further demonstrate changes in autophagic flux, we employed chloroquine and autophagic flux assays using a mCherry-GFP-LC3 construct in mPHs as previously described;[19] Red fluorescence, but not green fluorescence, is preserved in the acidic conditions of autolysosomes. AhR ablation facilitated autophagic degradation in adenoviral tandem mCherry-GFP-LC3 assays (Fig. 2h). Moreover, loss of AhR enhanced LC3-II level in the presence of chloroquine (Supplementary Fig. 2j, left and middle), whereas AhR overexpression exerted the opposite effect (Supplementary Fig. 2j, right). Consistently, a deficiency of AhR diminished the inhibitory effect of alcohol on autophagy (Supplementary Fig. 2k). These results provide strong evidence that alcohol-induced AhR overinduction and activation dysregulates autophagy flux in hepatocytes.

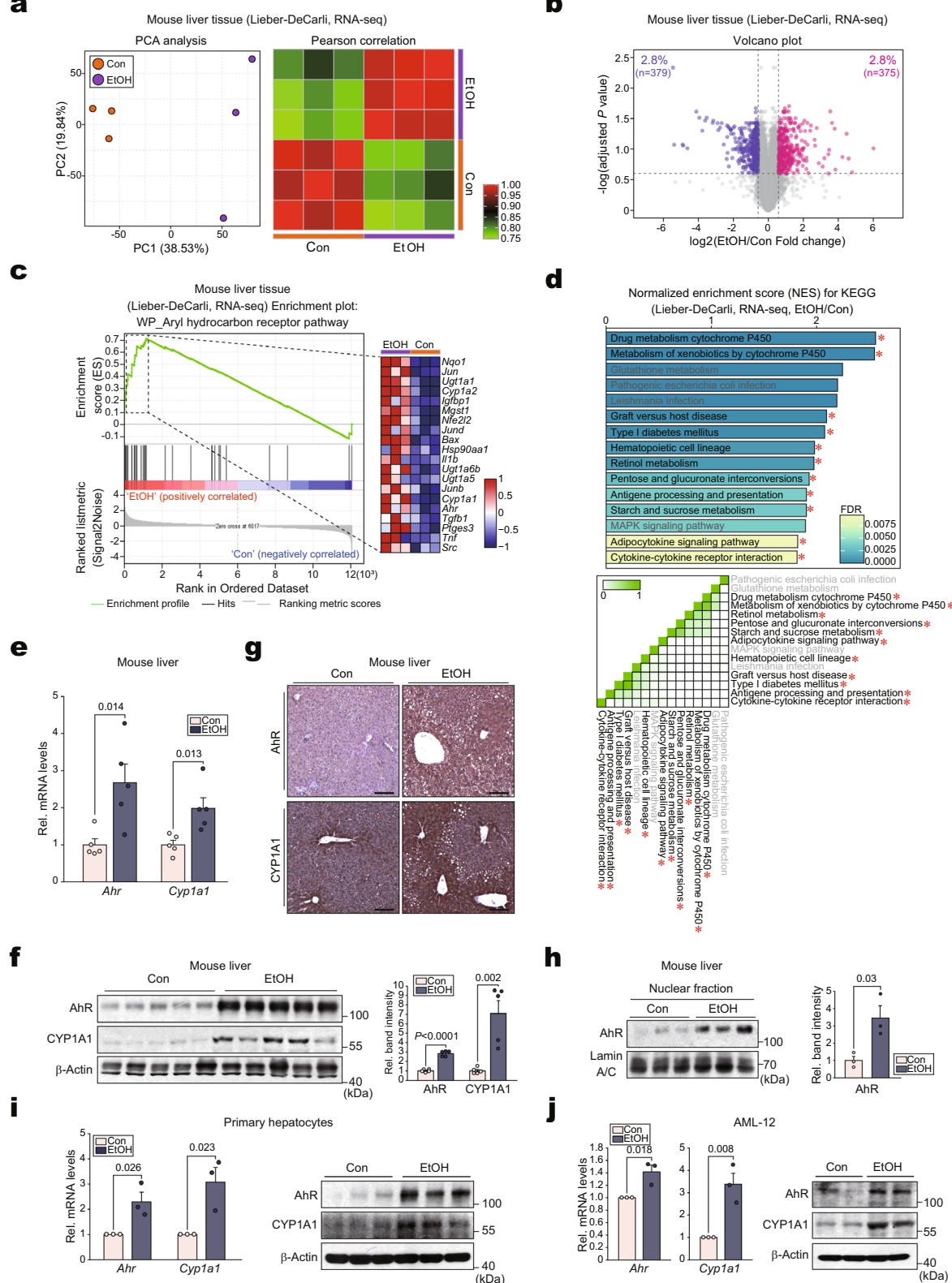

## Kynurenine produced by alcohol causes autophagy dysregulation through AhR

Although AhR is a well-studied nuclear receptor for xenobiotics[3], the physiological effect of its endogenous ligand has not been fully established. Therefore, we hypothesized that alcohol consumption affects kynurenine production through TDO2 and IDO. Our RNA-seq analysis indicated that exposure to the Lieber-DeCarli alcohol diet was correlated with the KEGG pathway of "Tryptophan metabolism" (Fig. 3a). Additionally, alcohol exposure promoted TDO2 and IDO expression in the liver (Fig. 3b–d). The direct effect of alcohol on the enzymes was confirmed using mPHs (Supplementary Fig. 3a).

Metabolomic profiling analyses were conducted to confirm the production of kynurenine upon alcohol exposure. The sparse partial least squares discriminant analysis (sPLS-DA) score plot indicated

**Fig. 1 | Activation of AhR in hepatocytes by alcohol. a** PCA score plot (left) and Pearson correlation matrix heatmap (right) of the hepatic transcriptome data obtained from mice fed with either a control diet or a Lieber-DeCarli alcohol liquid diet for 5 weeks (*n* = 3 each; darker green, closer negative correlation; darker red, closer positive correlation). **b** Volcano plot of DEGs using the same data as in (**a**) (blue, downregulated; red, upregulated; DEGs with adjusted *P* value <0.25 and absolute FC >1.5). **c** GSEA-enrichment plot of the AhR pathway (WikiPathways, WP) from the same data as in (**a**) (NES = 2.23, FDR <0.0001). The top 20 genes comprising the leading-edge of the enrichment score are shown in the corresponding heatmap (darker blue, stronger downregulation; darker red, stronger upregulation). **d** Bar graph (upper) and leading-edge analysis (lower) of the significantly enriched GSEA KEGG pathway using the same data as in (**a**). NES and FDR are presented as a bar graph (NES >1.81, FDR <0.01) (upper). The results from GSEA leading-edge analysis are shown as a similarity matrix where the intensity of the

green color directly correlates with the extent of the intersection between the leading-edge core genes of each gene set combination (lower). AhR-related pathways were marked with red asterisks. **e**–**g** qRT-PCR, immunoblotting, and immunohistochemical analyses for AhR and CYP1A1 in the liver of mice fed the Lieber-DeCarli diet for 4 weeks (*n* = 5 each). Representative images were shown for (**g**). Scale bar: 100 μm. **h** Immunoblot analyses for nuclear fractions prepared from liver samples of mice subjected to the Lieber-DeCarli diet for 4 weeks (left) and their quantifications (right) (*n* = 3 each). **i**, **j** qRT-PCR (left) or immunoblot (right) assays for AhR and CYP1A1 in mPHs isolated from the mice fed as indicated in (**a**) (**i**; *n* = 3 each). AML-12 cells were treated with 100 mM ethanol for 48 h (**j**; repeated three times with similar results). Values are expressed as means ± SEM. Significantly different compared to the controls. The data were analyzed via a two-sample two-tailed *t*-test with the Benjamini−Hochberg correction for an adjusted *P* value (**b**) or two-tailed Student's *t*-tests (**e**, **f**, **h**–**j**). Source data are provided as a Source Data file.

complete clustering between the two groups, suggesting that the metabolite profiles of the liver or serum samples obtained from mice fed with an ethanol diet were markedly different from those of the controls (Fig. 3e). A Pearson correlation matrix of all 41 amino acids and their metabolite levels in the liver also showed a distinct separation between the ethanol-exposed and control groups (Supplementary Fig. 3b). Among all amino acids and their derivatives, kynurenine was identified as the most significantly upregulated metabolite in liver and blood samples (Fig. 3f). This disparity between metabolomic profiles was largely attributed to changes in ten metabolites. Particularly, kynurenine exhibited higher VIP scores (Supplementary Fig. 3c; i.e., higher VIP scores indicate a greater kynurenine contribution). Therefore, ethanol exposure enhanced the liver and serum concentrations of kynurenine, as well as kynurenine to tryptophan ratios (Fig. 3g). Exposure of *Ahr* HKO mice to alcohol also increased kynurenine level in the liver, but weakly changed that of serum kynurenine (Supplementary Fig. 3d). This may be due to other factors such as altered metabolite transport, transporter expression, and kynurenine utilization. Among the top 20 most enriched pathways, we additionally recognized the metabolic pathways associated with tyrosine using Metabolite Set Enrichment Analysis (MSEA) (Supplementary Fig. 3e).

Having identified the inhibitory effect of AhR on autophagy flux, we hypothesized that AhR activation by kynurenine affects autophagy. As expected, autophagic flux was impaired after kynurenine treatment in mPHs or AML-12 cells, and the effect of AhR ablation on this event was verified by changes in LC3B and p62 (Fig. 3h, i). Similar results were obtained using FICZ, another endogenous AhR ligand (Supplementary Fig. 3f). Moreover, the inhibitory effect of kynurenine on autophagic flux was visualized using mPHs infected with adenoviral tandem mCherry-GFP-LC3. AhR ablation abrogated the effects of kynurenine on autophagic flux (Fig. 3j). Consistently, treatment with IDO/TDO2 inhibitors enhanced autophagy flux, which was attenuated by kynurenine (Supplementary Fig. 3g). Our findings suggest that increases in kynurenine through TDO2 and IDO contribute to AhR-dependent dysregulation of autophagy.

#### Alcohol inhibits AMPK via kynurenine-dependent AhR activation

We next explored the mechanisms by which kynurenine-mediated AhR activation dysregulates autophagy using RNA-seq analysis. PCA of the bulk gene expression levels indicated a significant separation of *Ahr* HKO from WT under ethanol exposure (Fig. 4a, left). Moreover, a Pearson correlation matrix was generated using all 13,345 gene expression profiles to compare the transcriptomes from each sample (Fig. 4a, right). The distinct gene expression profiles of the WT and *Ahr* HKO groups were visualized using a heatmap (Fig. 4b). The DEGs accounted for ~1% of the total transcriptome. Among the 127 DEGs, 55 genes were downregulated with 72 genes upregulated (Supplementary Fig. 4a). Additional pathways were analyzed using the newly elucidated

DEGs. Intriguingly, we identified upregulation of the "AMPK signaling pathway" (Fig. 4c).

AMPK plays a crucial role as a sensor of energy status, promoting the onset of autophagy[20,21]. Moreover, ethanol treatment inhibited AMPK activity[22], as also demonstrated by our experiments (Supplementary Fig. 4b). To assess whether AhR suppresses AMPK activity, we measured the level of p-AMPKα in the liver. p-AMPKα levels were lower in ethanol-fed mice; however, this effect was fully abrogated in *Ahr* HKO mice (Fig. 4d, e). Similar outcomes were obtained in the experiments using mPHs isolated from mice fed with an alcohol diet (Fig. 4f) or mPHs and AML-12 cells treated with alcohol or kynurenine after AhR deletion in vitro (Fig. 4g, h). Additionally, treatment with kynurenine and FICZ inhibited AMPK (Supplementary Fig. 4c), being consistent with the outcomes of alcohol treatment and AhR overexpression (Supplementary Fig. 4d). As expected, chemical modulations of AMPK activity appropriately changed p-ULK1 levels in the cells treated with alcohol (Supplementary Fig. 4e), as shown previously[23,24]. Likewise, alcohol treatment decreased p-ULK1 level, which was reversed by AhR deficiency (Supplementary Fig. 4f). These data confirmed that AhR-dependent inhibition of AMPK decreases ULK1 activation. Altogether, our results strongly suggest that kynurenine-mediated activation of AhR leads to AMPK inhibition upon alcohol exposure.

#### Transcriptional induction of *Ppp2r2d* by AhR dysregulates autophagy through AMPK inhibition

To elucidate the molecular mechanisms by which ligand-activated AhR inhibits p-AMPKα, we considered LKB1 as a potential target of AhR but found no effect of AhR on LKB1 (Supplementary Fig. 5a). We thus focused on the AMPK phosphatase. As shown in Supplementary Fig. 5b, PP2A, a heterotrimeric serine/threonine phosphatase consisting of subunits A, B, and C, accounted for most of the serine/threonine phosphatase activity[25]. Particularly, PPP2R2D, PPP2R5C, and PPP2R5D control AMPK phosphorylation by dephosphorylating at T172[26,27]. Therefore, we examined *Ppp2r2d*, *Ppp2r5c*, and *Ppp2r5d* in the liver of alcohol-fed mice and found that *Ppp2r2d*, but not the other phosphatases, was upregulated (Fig. 5a). The ability of alcohol to induce PPP2R2D was confirmed by immunoblotting and immunohistochemistry (Fig. 5b). Further, the direct effect of alcohol on PPP2R2D was confirmed in AML-12 cells (Fig. 5c).

We next assessed the role of AhR in the alcohol-mediated induction of PPP2R2D using knockout and knockdown models. *Ahr* HKO prevented alcohol from increasing PPP2R2D mRNA and protein levels in the liver or mPHs prepared ex vivo from mice fed with a Lieber-DeCarli diet for 5 weeks (Fig. 5d–g). AhR modulation using gene deletion, small-interfering RNA (siAhR), or overexpression approaches resulted in the expected outcomes in vitro (Fig. 5h and Supplementary Fig. 5c), confirming the key role of AhR in regulating PPP2R2D expression.

Further, we confirmed the effect of kynurenine on the expression of PPP2R2D, confirming that kynurenine treatment

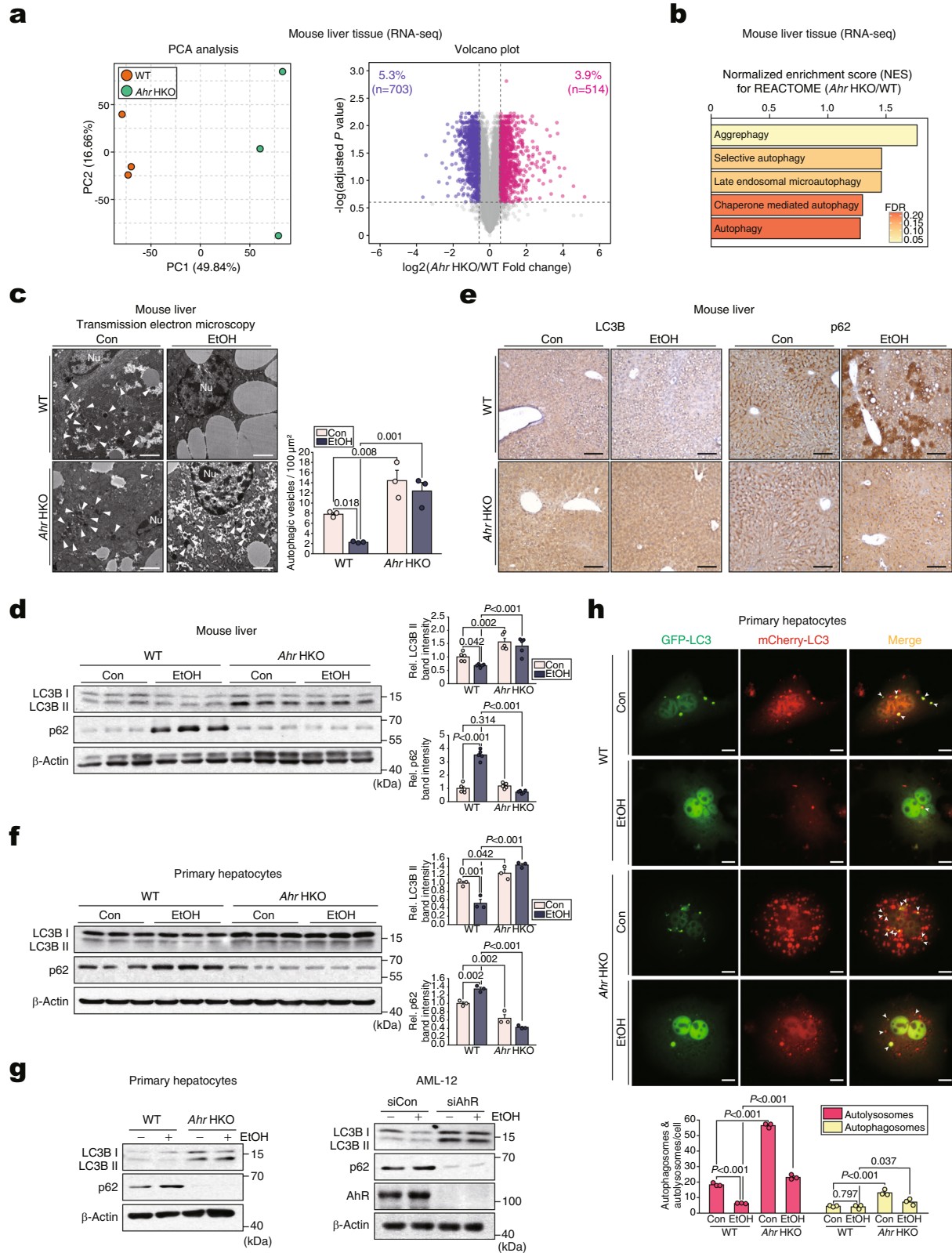

facilitated the induction of PPP2R2D in AML-12 cells (Fig. 5i). Similar outcomes were obtained using FICZ (Supplementary Fig. 5d). The key role of AhR in the induction of PPP2R2D by kynurenine was verified using *AhR* knockout and knockdown methods in cells (Fig. 5j).

The JASPAR database was used to validate the transactivating effect of AhR on the target gene, and two putative AhR responsive elements were found within the −2 kb promoter region of *Ppp2r2d* (Fig. 5k, left). Chromatin immunoprecipitation (ChIP) and qRT-PCR assays confirmed that AhR interacted with the DNA elements (Fig. 5k, right). Luciferase reporter assays were used to validate the effect of AhR to transactivate *Ppp2r2d*. As expected, AhR-mediated luciferase expression was abrogated in the assays using a mutant construct, confirming AhR dependency (Fig. 5l).

**Fig. 2 | Augmentation of autophagy flux by hepatocyte-specific *Ahr* deficiency in mice. a** PCA score plot (left) and volcano plot (right; blue, downregulation; red, upregulation) of DEGs based on the hepatic transcriptome data using WT or *Ahr* HKO mice (*n* = 3 each; DEGs of adjusted *P* value <0.05 and absolute FC >1.5). **b** Autophagy-related pathways based on GSEA REACTOME analysis using the same data as in (**a**). FDR and NES are presented as a bar graph (NES >1.25, FDR <0.25). **c** Representative TEM (Nu, nucleus) in the liver of mice subjected to a control diet or Lieber-DeCarli diet for 5 weeks (left) and their quantifications (right) (*n* = 3 each). The arrows indicate autophagic vesicles containing organelle remnants. Scale bar: 2 μm. **d** Immunoblots of the liver homogenates of mice fed as in (**c**) (left) and their quantifications (right) (*n* = 5 each). **e** Representative immunohistochemical images for LC3B (left) and p62 (right) using the same mice as in (**d**) (*n* = 5 each). Scale bar: 100 μm. **f** Immunoblots in mPHs isolated from the mice fed as in (**c**) (left) and their quantifications (right) (*n* = 3 each). **g** Immunoblots for LC3BI/II and p62 in the lysates of mPHs treated with 100 mM ethanol for 48 h (left) or AML-12 cells treated with 100 mM ethanol for 48 h after transfection with siAhR (or siCon) for 24 h (right) (repeated three times with similar results). **h** Representative confocal microscopic images of mCherry/GFP-LC3B puncta staining (upper) and their quantifications (lower) (*n* = 3 each, 10 cells/group for each experiment). To collect confocal images of yellow (autophagosomes) or red (autolysosome) puncta, mPHs were infected with Ad-mCherry-GFP-LC3B for 12 h and then exposed to 100 mM ethanol (or Con) for 48 h. Scale bar: 10 μm. Values are expressed as means ± SEM. Significantly different compared to the control or WT. The data were analyzed via a two-sample two-tailed *t*-test with the Benjamini–Hochberg correction for an adjusted *P* value (**a**) or one-way ANOVA with LSD (**c**, **d**, **f**, **h**). Source data are provided as a Source Data file.

Having identified the induction of *Ppp2r2d* by AhR, we next examined the role of PPP2R2D in alcohol-induced AMPK phosphorylation and autophagy dysregulation. PPP2R2D knockdown abrogated the inhibitory effect of alcohol on AMPK in mPHs or AML-12 cells (Fig. 5m, left and Supplementary Fig. 5e, left). Similar outcomes were obtained in the experiment using kynurenine (Fig. 5m, right and Supplementary Fig. 5e, right). In contrast, PPP2R2D overexpression yielded the opposite results (Supplementary Fig. 5f). Moreover, autophagy was suppressed by PPP2R2D in hepatocytes, as indicated by changes in LC3B cleavage and p62 expression (Fig. 5n and Supplementary Fig. 5g, h). In the assays using adenoviral mCherry-GFP-LC3, the inhibition of autophagy flux in mPHs by either alcohol or kynurenine was notably attenuated by siPPP2R2D transfection (Supplementary Fig. 5i). By the same token, treatment with IDO/TDO2 inhibitors diminished ethanol effects on the identified targets in the experiment using mPHs isolated from mice fed with an alcohol diet (Supplementary Fig. 5j). Together, our results indicate that kynurenine-mediated activation of AhR enhanced by alcohol promotes transcriptional induction of *Ppp2r2d*, and thus autophagy might be dysregulated by the resulting AMPK inhibition.

## AhR ablation improves fat catabolism and alcoholic fatty liver via mitochondrial fuel consumption

Alcohol consumption causes fat accumulation in hepatocytes and liver damage[5], as verified by our experiments (Supplementary Fig. 6a–d). To address the impact of AhR activation on alcoholic fatty liver, we examined the hepatic fat contents in WT and *Ahr* HKO mice fed with a Lieber-DeCarli alcohol diet. AhR ablation almost entirely prevented excess neutral fat accumulation, as indicated by changes in histopathology, Oil Red O staining, and hepatic or serum triglyceride (TG) contents (Fig. 6a, b). Alcohol treatment to *Ahr* HKO mice did not change the liver weight compared to WT control, but alcohol-induced increases in the liver to BW ratios were attenuated in *Ahr* HKO mice (Supplementary Fig. 6e) and the level of liver injury markers in blood returned to normal (Supplementary Fig. 6f).

Based on the known effect of autophagy dysregulation on fat accumulation coupled with our findings regarding the regulatory role of AhR in lipid metabolism[28], we next performed lipidomic analysis and found a distinct separation of lipid metabolites composition among the WT, WT + alcohol, and *Ahr* HKO + alcohol groups, as demonstrated by our PCA results (Fig. 6c). The lipidomics heatmap indicated that alcohol treatments clearly increased the levels of C28-C44 saturated/unsaturated phosphatidylcholines; and of sphingolipids including saturated/unsaturated C14-C22 ceramides and saturated/unsaturated C16-C22 sphingomyelins, which comprised 60% among the lipids significantly affected. AhR ablation prevented alcohol from increasing most of the phosphatidylcholines and sphingolipids levels as well as those of diacyl glycerides (Fig. 6d). Moreover, *Ahr* HKO enhanced C14-C20 saturated/unsaturated lysophosphatidylcholines, suggestive of phosphatidylcholine chain cleavage and the resultant fatty acid

utilization. RNA-seq assays were further conducted using the same diet models. Notably, "Obesity up" was the most highly upregulated among the NADLER pathways (Supplementary Fig. 6g). In BODIPY staining assays, ablation of AhR decreased lipid droplets under alcohol conditions, which was reversed by chloroquine treatment (Supplementary Fig. 6h). Collectively, these results suggest that AhR deficiency prevents alcohol-induced disruption of lipid metabolism via autophagy regulation.

We had previously demonstrated a link between autophagy-dependent lipid catabolism and mitochondrial function[29]. Therefore, pathway analyses were conducted using an RNA-seq dataset. Among the top 15 KEGG pathways, AhR ablation upregulated the pathways related to mitochondrial function, NADH dehydrogenase assembly, and electron transport (Fig. 6e and Supplementary Fig. 6i), being consistent with the changes in phosphatidylcholines and sphingolipids contents. Likewise, alcohol treatment significantly decreased mitochondrial DNA contents, which was reversed by AhR deficiency (Fig. 6f). We also measured the levels of transcripts encoding for mitochondrial biogenesis- and respiratory chain-associated proteins. *Ahr* HKO enhanced all of the mRNAs in mPHs (Fig. 6g). Finally, we monitored hepatic lipid metabolism and mitochondrial function by flow cytometry using Nile red and Rhodamine 123[30,31]. AhR overexpression increased Nile red staining intensity while decreasing Rhodamine fluorescence (Supplementary Fig. 6j). Therefore, our data indicated that AhR promotes excess lipid accumulation and mitochondrial dysfunction.

Fuel production by autophagy is linked to mitochondrial oxygen consumption[14]. Finally, to understand the roles of AhR and PPP2R2D on mitochondrial fuel oxidation, we assessed oxygen consumption rates (OCRs) in mPHs. OCRs were significantly increased in mPHs isolated from *Ahr* HKO mice compared with WT (Fig. 6h, upper). Alcohol exposure diminished mitochondrial OCRs in mPHs (Supplementary Fig. 6k), being consistent with previous findings that ethanol inhibits mitochondrial respiration[32]. As expected, AhR ablation prevented alcohol from decreasing OCRs (Fig. 6h, lower). Moreover, chloroquine treatment completely reversed the OCR-promoting effect of AhR ablation (Fig. 6i), suggesting autophagy-mediated beneficial effects of AhR deficiency. Moreover, knockdown of PPP2R2D promoted mitochondrial OCRs (Fig. 6j, upper) and this effect was also observed in alcohol-exposed mPHs (Fig. 6j, lower). Additionally, PPP2R2D transfection abrogated the OCR-promoting effect of *Ahr* HKO (Fig. 6k), confirming that AhR-mediated regulation of OCRs in mitochondria is largely dependent on PPP2R2D and autophagy.

To validate the outcomes in a more severe alcohol model, we employed an NIAAA (chronic-binge feeding) mouse model, and confirmed similar changes in AhR, CYP1A1, IDO, and TDO2 levels (Supplementary Fig. 7a–c). Ablation of AhR also blunted the basal and alcohol-inducible expression of PPP2R2D with appropriate changes in p-AMPK and autophagy marker levels (Supplementary Fig. 7d–f). Consistently, deletion of AhR in hepatocytes not only

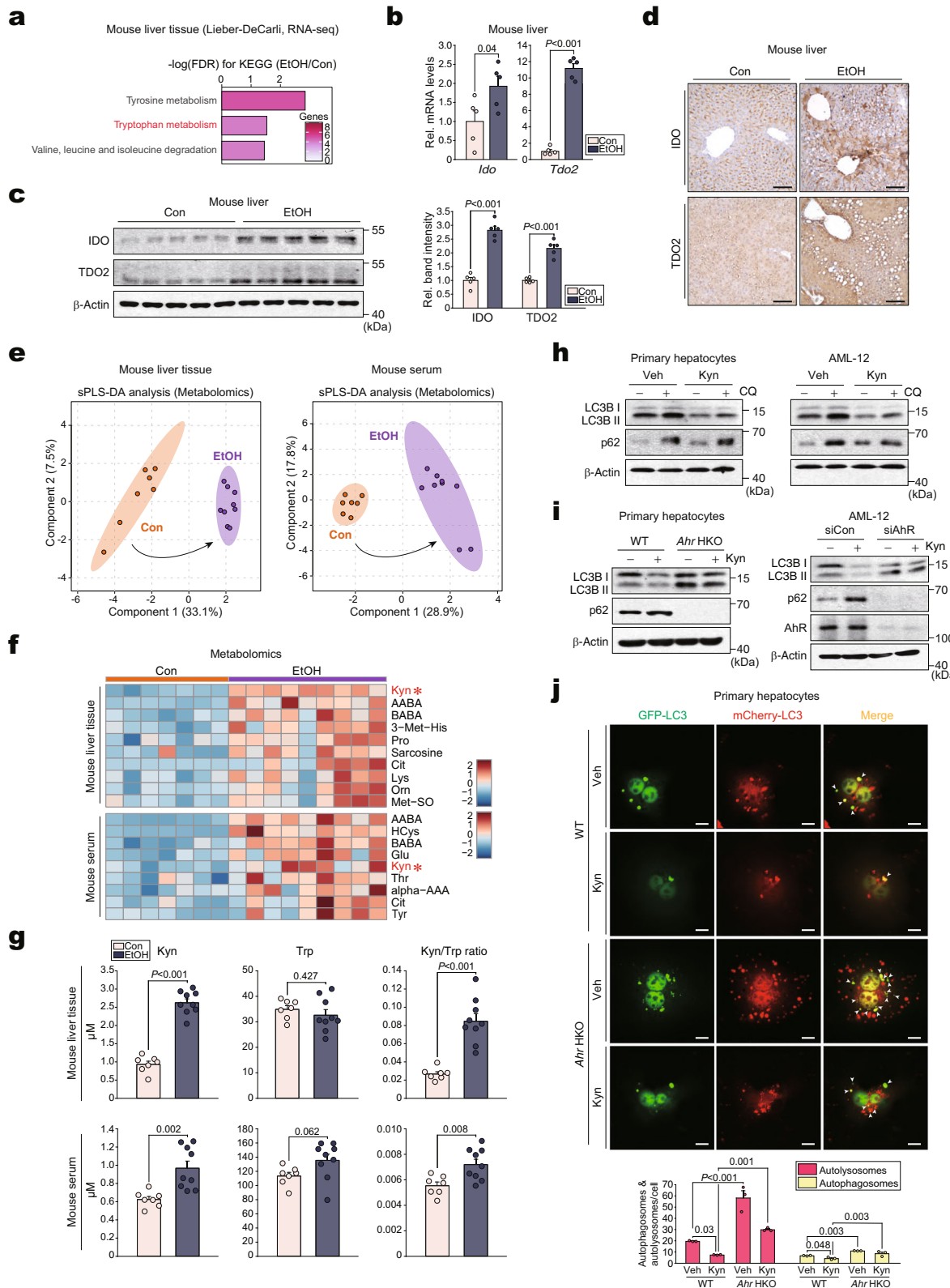

amerliorated lipid accumulation elicited by chronic-binge alcohol feeding, but also improved histopathological features (Supplementary Fig. 7g, h). Likewise, we confirmed changes in the liver to BW ratios and liver injury blood markers (Supplementary Fig. 7i, j). Altogether, these findings strengthen the functional role of AhR in regulating the PPP2R2D-AMPK axis and the resultant autophagy pathway in alcoholic liver disease.

## The identified AhR targets correlate with autophagy markers in patients

Finally, we analyzed liver specimens from patients with ALD. Histological analysis of the liver showed an increase in hepatic steatosis in ALD patients compared to the control group without ALD (Supplementary Fig. 8a). Additionally, *CYP1A1*, *PPP2R2D*, and *IDO* transcript levels were elevated in ALD patients (Fig. 7a). *CYP1A1* transcript levels were

**Fig. 3 | Production of kynurenine by alcohol and its effect on autophagy flux.**
**a** Bar graph based on KEGG analysis to identify functional processes controlled by alcohol treatment using the same data as in Fig. 1a. FDR and genes are presented as a bar graph (FDR <0.05). **b**–**d** qRT-PCR, immunoblots, and immunohistochemical analyses for IDO and TDO2 in the liver of mice fed with the control diet or a Lieber-DeCarli alcohol liquid diet for 4 weeks ($n = 5$ each). Representative images were shown for (**d**). Scale bar: 100 μm. **e** sPLS-DA score plots based on the metabolomics data related to amino acids using the liver (left) and serum (right) from the mice fed either a control diet or a Lieber-DeCarli diet for 5 weeks ($n = 7$, 9 mice). **f** Heatmap analyses of increased amino acids and their metabolites using the same data as in (**e**) ($n = 7$, 9 mice; FDR <0.05; darker blue, stronger downregulation; darker red, stronger upregulation). Kynurenine was marked with red asterisks.
**g** Concentrations of kynurenine and tryptophan, and kynurenine/tryptophan ratios

using the same data as in (**e**) ($n = 7$, 9 mice). **h** Immunoblots for LC3BI/II and p62 in mPHs (left) or AML-12 cells (right) treated with 100 μM kynurenine for 12 h and continuously with 10 μM CQ for 2 h (repeated three times with similar results). **i** Immunoblots for LC3BI/II and p62 in mPHs treated with 100 μM kynurenine for 12 h (left) or AML-12 cells treated with 100 μM kynurenine for 12 h after transfection with siAhR (or siCon) for 24 h (right) (repeated three times with similar results). **j** Representative confocal microscopic images of staining for mCherry/GFP-LC3B puncta (upper) and their quantifications (lower) ($n = 3$ each, 10 cells/group for each experiment). mPHs were infected with Ad-mCherry-GFP-LC3B for 12 h and then exposed to 100 μM kynurenine (or vehicle) for 12 h. Scale bar: 10 μm. Values are expressed as means ± SEM. Significantly different compared to Con, Veh, or WT. Data were analyzed via two-tailed Student's t-test (**b**, **c**, **g**) or one-way ANOVA with LSD (**j**). Source data are provided as a Source Data file.

positively correlated with *PPP2R2D* (Fig. 7b). Immunoblotting analyses further confirmed the changes in the corresponding gene products along with decreases of p-AMPKα and autophagy markers in the patients (Fig. 7c). Further, the levels of p62 and IDO, but not TDO2, were increased. The outcomes were confirmed via immunohistochemistry (Fig. 7d). Additionally, AhR intensities were positively correlated with PPP2R2D in the patients, whereas *CYP1A1* and p-AMPKα levels were inversely correlated (Supplementary Fig. 8b). Likewise, CYP1A1 and PPP2R2D were negatively associated with autophagy (Fig. 7e).

Furthermore, AhR, CYP1A1, and *PPP2R2D* expression in ALD patients increased with decreases in p-AMPKα as histological steatosis scores worsened (Fig. 7f). Similar results were obtained depending on fibrosis stages (Supplementary Fig. 8c). Additionally, plasma γ-glutamyl transferase (GGT) activities were considerably higher in ALD patients than in control without ALD participants ($303.5 \pm 50.71$ vs. $50.63 \pm 19.74$ IU/L, $P = 0.001$) (Supplementary Table 1), and the identified target levels (IDO, AhR, *PPP2R2D*, and p-AMPKα/AMPKα) were significantly correlated with those of GGT (Supplementary Fig. 8d). Therefore, we concluded that AhR dysregulates autophagy flux in ALD patients through the PPP2R2D-AMPK axis, thereby exacerbating hepatic lipid accumulation and hepatocyte injury (Fig. 7g).

## Discussion

In this study, we demonstrate that the metabotropic activation of AhR is mediated by kynurenine, a metabolite generated by IDO and TDO, and is affected by repetitive alcohol treatments. The outcomes of RNA-seq analyses and in vivo or in vitro-based approaches clearly show that alcohol intake causes kynurenine-dependent overexpression of AhR. Our analyses also confirmed significant increases of kynurenine in both the liver and serum and activation of the AhR signaling network, supporting that increase of kynurenine by alcohol stimulates AhR and its downstream cascades in hepatocytes. The transcriptomic and metabolomic analyses thus enabled us to extract a cluster of genes associated with lipid metabolism, as confirmed by changes in the experimental lipidomics assays. The AhR acts through AHRE DNA elements, activating or repressing target genes, which include CYP1A1 and Nrf2[33,34]. In contrast to our finding showing an AhR-mediated deleterious effect of alcohol, others insisted that high productions of bacterial indoles, as tryptophan metabolites, might be associated with improvement of hepatic injury[9,35]. So, there exist differential roles of AhR in ALD progression, especially in association with environmental factors such as pollutants affecting hepatic enzymes responsible for ethanol and/or modification of microbiomes[36,37]. Admittedly, the gut-liver axis plays a role in driving the disease[38]. As the gut microbiome is a source of endogenous AhR ligands, the produced AhR ligands may reach the liver via portal blood, affecting liver biology[39]. In mammals, however, >90% of tryptophan is degraded through the hepatic kynurenine pathway[40,41]. So, the liver may play a major role in tryptophan catabolism so that excessively increased kynurenine and the

resultant AhR activation in the liver may obscure the intestinal effect. Of the kynurenine-producing enzymes, IDO was shown to be induced by IFN-γ, a cytokine linked to alcohol consumption, in many types of cells, including hepatocytes[42–44]. In other studies, ethanol treatment elevated plasma glucocorticoid and TDO2 activity in the liver[45,46]. So, it seems that IDO and TDO2 levels are regulated through different mechanisms.

Another important finding of our study is that AhR induction and its metabotropic activation by kynurenine causes autophagy inhibition, as strengthened by the results of experiments using *Ahr* HKO animal and hepatocyte models. Moreover, the outcomes of our RNA-seq analyses corroborated the functional roles of AhR and autophagy in hepatocytes in steatosis progression leading to liver dysfunction. These physiological effects of AhR after endogenous ligand binding and its impact on hepatic lipid accumulation are of particular interest in terms of potential therapeutic intervention. The RNA-seq and in vivo/in vitro-based approaches enabled us to identify AMPK as a signaling node for autophagy downstream from AhR, being consistent with our other finding that AhR deficiency prevents alcohol from inhibiting kynurenine activation of AMPK. The autophagic flux can be influenced by the early-stage or late-stage of autophagy via alcohol, although it is still complex and contentious in ALD; Some studies showed elevated levels of LC3 and p62 due to dysfunction of lysosomes[7,47], whereas others and ours exhibited LC3 decrease in vivo or in vitro[48,49]. In the present study, we confirmed the regulatory roles of AhR and AMPK in p-ULK1, a central player in the early stage of autophagy[23,24]. All together, we carefully surmise that AhR may play a role in the early stage of autophagy.

Phosphatase may be an important target for drug discovery, as in the case of protein phosphatase 2B inhibitors[50,51]. Our results provide evidence of a mechanism wherein AhR induces PPP2R2D, elucidating the role of PPP2R2D in the dephosphorylation of p-AMPKα, which suppresses AMPK-mediated autophagy; consequently, alcohol-induced kynurenine activation of AhR inhibits autophagic flux through the PPP2R2D-AMPK axis. Hence, the AhR-PPP2R2D pathway discovered herein has implications for both hepatic lipid metabolism and mitochondrial function in response to alcohol intake, providing insights into the upstream regulators of alcohol dysregulation of autophagy. Moreover, metabotropic activation of AhR by alcohol led to dysregulation of phospho-/sphingo-lipids metabolism, presumably affecting cell membrane biology and signaling.

Notably, hepatic mitochondrial oxidative function encompasses crucial pathways, including β-oxidation and the respiratory chain, both of which are closely linked to lipid metabolism[52]. Additionally, autophagy inhibitors impair mitochondrial function, as mitochondria are also intimately linked to the process of autophagy[53]. Consequently, alcohol treatments caused drastic alterations in phospho-/sphingo-lipids levels in the liver through AhR, and thus AhR ablation enhanced lysophosphatides, indicative of fat catabolism. The disruption of phospho-/sphingo-lipids by alcohol may also account for liver dysfunction, as indicated by a decrease in mitochondrial respiration along

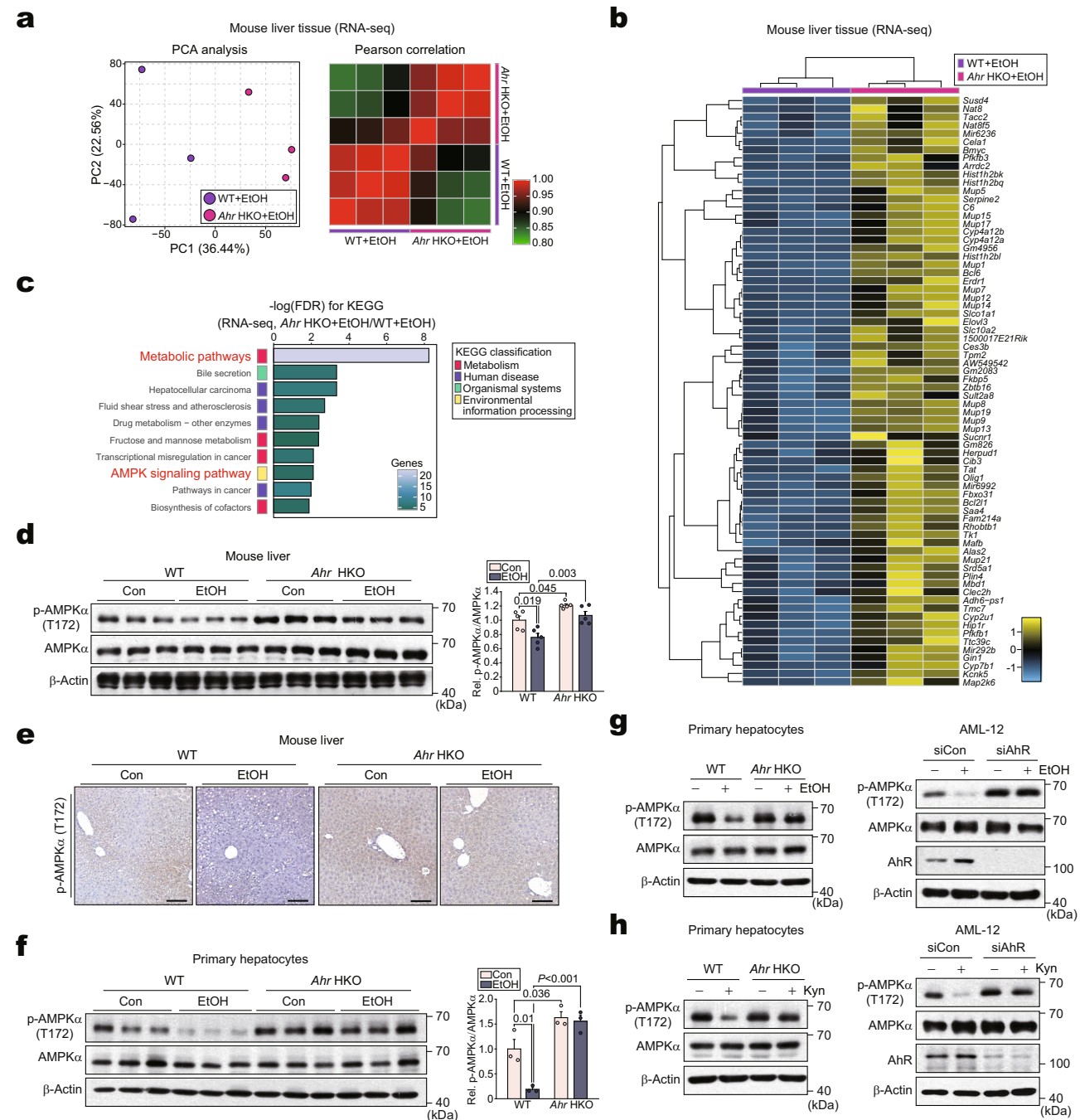

**Fig. 4 | AhR-dependent inhibition of p-AMPKα in mice or hepatocytes treated with alcohol. a** PCA score plot (left) and heatmap of the Pearson correlation matrix (right) based on the hepatic transcriptome data obtained from WT or *Ahr* HKO mice fed Lieber-DeCarli diet for 5 weeks (*n* = 3 each; darker green, closer negative correlation; darker red, closer positive correlation). **b** Heatmap and hierarchical correlation analyses of DEGs using the same data as in (**a**) (DEGs of *P* < 0.05 and FC >2). The DEGs were hierarchically clustered and presented as heatmap according to the row *Z*-score (darker blue, stronger downregulation; darker yellow, stronger upregulation). **c** Bar graphs based on KEGG analysis to identify functional processes controlled by *Ahr* HKO using the same data as in (**a**). FDR and genes are presented as bar graphs (FDR <0.01). **d** Immunoblots from the liver homogenates obtained from the mice subjected to control or Lieber-DeCarli diets for 5 weeks (left) and

their quantifications (right) (*n* = 5 each). **e** Representative immunohistochemical images for p-AMPKα using the same mice as in (**d**) (*n* = 5 each). Scale bar: 100 μm. **f–h** Immunoblots in mPHs isolated from the mice fed as in (**d**) (left) and their quantifications (right) (**f**; *n* = 3 each); mPHs treated with 100 mM ethanol for 48 h (left) or AML-12 cells treated with 100 mM ethanol for 48 h after transfection with siAhR (or siCon) for 24 h (right) (**g**; repeated three times with similar results); mPHs treated with 100 μM kynurenine for 12 h (left) or AML-12 cells treated with 100 μM kynurenine for 12 h after transfection with siAhR (or siCon) for 24 h (right) (**h**; repeated three times with similar results). Values are expressed as means ± SEM. Significantly different compared to Con or WT. Data were analyzed via one-way ANOVA with Tukey HSD (**d**, **f**). Source data are provided as a Source Data file.

with the suppression of genes encoding for mitochondrial biogenesis and oxidative phosphorylation, which returned to normal after *Ahr* deletion. Particularly, either an autophagy inhibitor or PPP2R2D overexpression reversed the OCR-promoting effects of AhR deletion,

corroborating the autophagy-dependent role of AhR and PPP2R2D in OCRs.

Past studies have shown that AhR plays a role in the formation and decomposition of lipids[54,55]. TFEB controls the biogenesis of

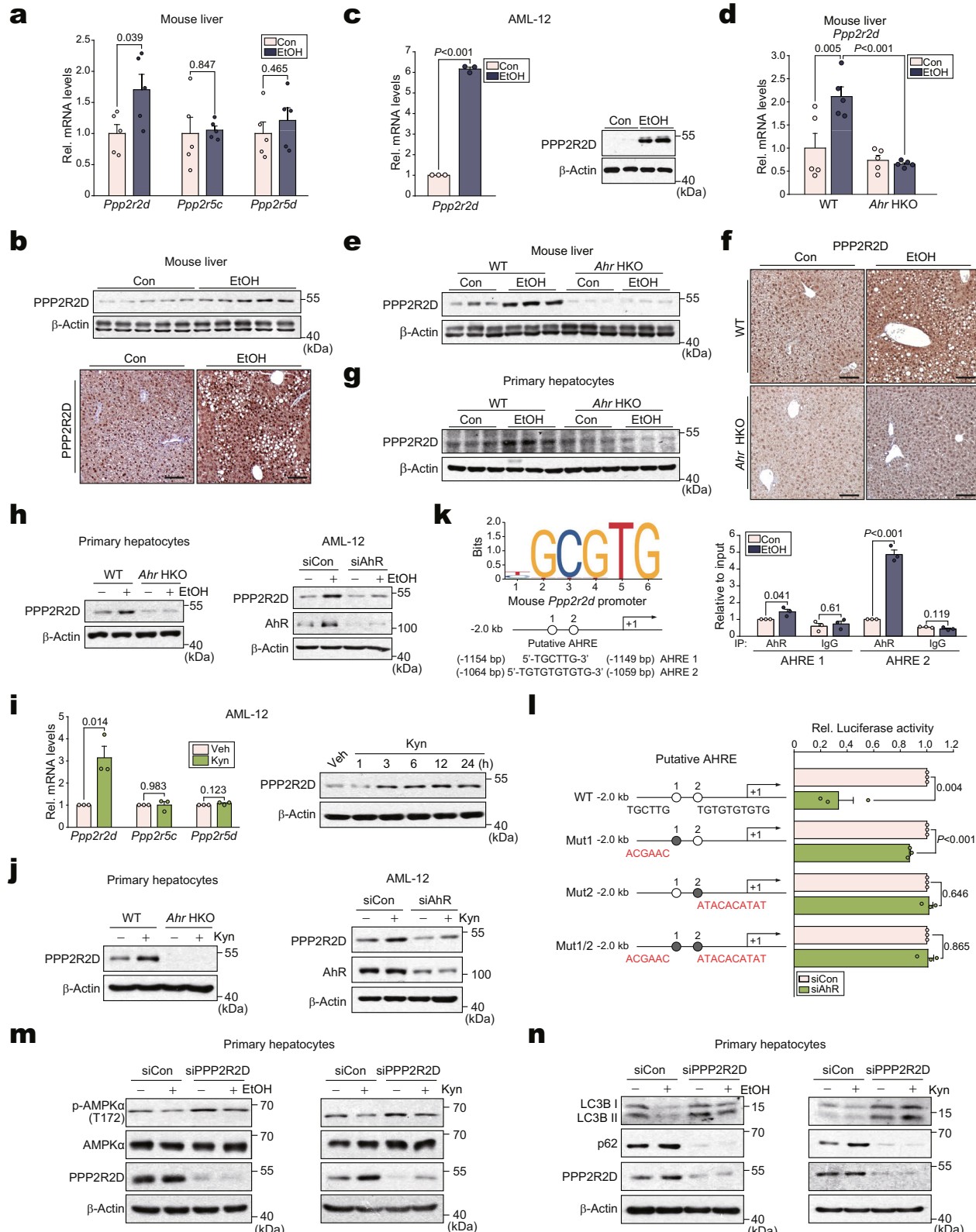

mitochondria and lipid metabolism by promoting PGC1α[56] and is transcriptionally regulated by PPARα[57]. Of the many target genes, AhR inhibits hepatic PPARα[58]. Thus, the TFEB pathway might be one of the possible mechanisms underlying the effect of AhR on mitochondria and lipid metabolism in alcohol conditions. AMPK plays a role as a sensor of energy status[20,21], and thus promotes de novo mitochondrial biogenesis while suppressing lipid accumulation in part by an

autophagy-independent manner[23,24]. Given these, in addition to the present finding that AhR interrupts autophagy flux by inhibiting AMPK signaling, we cannot rule out that other autophagy-independent pathways regulated by AhR/AMPK account for ALD improvement.

Our results using the Lieber-DeCarli diet model may have a limitation to understand the pathogenesis of alcoholic hepatitis in clinical situations. Nevertheless, this model and variations mimic different

**Fig. 5 | Suppression of autophagy by dephosphorylation of AMPKα through transcriptional induction of *Ppp2r2d* by AhR. a** qRT-PCR assays for isoforms of PP2A subunit B in the liver of mice fed with Lieber-DeCarli diets for 4 weeks ($n = 5$ each). **b** Immunoblot (upper) or representative immunohistochemical images (lower) for PPP2R2D in the same liver samples as in (**a**) ($n = 5$ each). Scale bar: 100 µm. **c** qRT-PCR (left) or immunoblot (right) assays for PPP2R2D in AML-12 cells (100 mM ethanol, 48 h; repeated three times with similar results). **d–f** qRT-PCR, immunoblot, and immunohistochemical analyses for PPP2R2D (Lieber-DeCarli diet for 5 weeks; $n = 5$ each). Representative images were shown for (**f**). Scale bar: 100 µm. **g** Immunoblot of mPHs isolated from the mice fed as in (**d**) ($n = 3$ each). **h** Immunoblots of PPP2R2D in mPHs (100 mM ethanol, 48 h) or AML-12 cells (siAhR transfection, 48 h; 100 mM ethanol, 24 h) (repeated three times with similar results). **i** qRT-PCR assays (left) or immunoblot (right) for PPP2R2D in AML-12 cells treated with 100 µM kynurenine for 12 h or the indicated times (repeated three

times with similar results). **j** Immunoblots for PPP2R2D in mPHs (100 µM kynurenine, 12 h) or AML-12 cells (transfection with siAhR, 24 h; 100 µM kynurenine, 12 h) (repeated three times with similar results). **k** ChIP assays using AhR immunoprecipitates from AML-12 cells (100 mM ethanol for 24 h; IgG immunoprecipitation, a negative control). One-tenth of cross-linked lysates served as the input control ($n = 3$ each). **l** *Ppp2r2d* promoter-reporter assays using WT or Mut reporters in AML-12 cells transfected with siAhR for 24 h ($n = 3$ each). **m** Immunoblots for p-AMPKα in mPHs treated with 100 mM ethanol for 48 h (left) or with 100 µM kynurenine for 12 h (right) after transfection with siPPP2R2D for 24 h (repeated three times with similar results). **n** Immunoblots for autophagy markers using the same samples as in (**m**) (repeated three times with similar results). Values are expressed as means ± SEM. Significantly different compared to Con, WT, Veh, or siCon. Data were analyzed via two-tailed Student's *t*-test (**a**, **c**, **i**, **k**, **l**) or one-way ANOVA with Tukey HSD (**d**). Source data are provided as a Source Data file.

medical settings of fatty liver and associated hepatic alterations[59,60]. Indeed, our results and others show that mice develop a moderate degree of fat accumulation, accompanied by inflammatory responses within one month of alcohol feeding, accounting for certain aspects of the human disease[61,62]. Consistently, the outcomes of this study confirmed the identified targets in patient liver specimens. In addition, our data using the primary hepatocyte culture system supports ethanol-induced hepatic injury; Nonetheless, the outcomes have limitations in understanding ALD-associated cell death and inflammation because of its mild effect.

The expression of CYP1A1, PPP2R2D, IDO, and autophagy flux in human samples were similar to those found in mice, but not AhR or TDO2. Although a substantial increase in the protein level was observed, mRNA levels remained largely unchanged, unlike in mice. This phenomenon could be explained by post-translational modifications of AhR (e.g., ubiquitination and SUMOylation)[63,64], yet additional studies are required to confirm this notion. In the present study, TDO2 mRNA and protein levels remained largely unchanged in the patients. However, in some human studies, alcohol consumption increased serum and urine kynurenine levels[65–67], being in line with our hypothesis. Given that the affinity of IDO ($Km = 20$ µM) is five times higher than that of TDO2 ($Km = 100$ µM)[68], it is very likely that the effects of alcohol on kynurenine are mainly due to an increase in IDO level. Moreover, both the steatosis score and GGT significantly correlated with the expression of targets, suggesting that these proteins are associated with lipid accumulation and liver injury.

In summary, we identified the previously uncharacterized role of AhR in interrupting hepatic autophagy via AMPK inhibition, and elucidated PPP2R2D as a mediator of this effect. Our findings provide strong evidence that AhR in hepatocytes activated by kynurenine promotes disease progression by inducing PPP2R2D and impeding AMPK signaling. Further, our results show that hepatocyte-specific deletion of *Ahr* attenuates defective autophagy, hepatic steatosis, and mitochondrial dysfunction with the homeostatic recovery of phospho-/sphingo-lipids levels. These findings provide the concept that hepatocyte-specific inhibition of *Ahr* and its downstream PPP2R2D may hold promise for the treatment of patients suffering from alcohol in the context of autophagy regulation.

## Methods

### Analysis of liver specimens from ALD patients
Liver biopsy samples were obtained from patients with ALD. Written informed consent was obtained from all participants before participating in the research studies. There was no compensation for participants. Studies using human samples were reviewed and approved by the independent Institutional Review Board of the Seoul Metropolitan Government Seoul National University Boramae Medical Center (Seoul, South Korea). The study conformed to the ethical guidelines of the World Medical Association Declaration of Helsinki and was approved by the Institutional Review Board of Seoul Metropolitan

Government Seoul National University Boramae Medical Center (IRB No. 16-2013-45). The clinical characteristics of the patients are listed in Supplementary Tables 1–3. Fresh liver tissue samples were obtained from some of the patients with ALD and control (without ALD) were analyzed by qRT-PCR, western blot, and immunostainings. The patients were subdivided and analyzed according to the steatosis score. H-score was calculated using the ImageJ (version 1.53p, NIH image program) software and the IHC profiler plugin. To determine AhR, CYP1A1, PPP2R2D, p-AMPKα, LC3B, p62, IDO, and TDO2 expression in ALD patients, the standard protocol designed by ref. 69 was followed. *H*-score was assigned using the formula ($1 \times$ (%cells low positive) + $2 \times$ (%cells positive) + $3 \times$ (%cells high positive)), obtaining a value from 0 to 300.

### Animals
The mice were housed at $22 \pm 2$ °C with a 12-h light/12-h dark cycle and relative humidity of $55 \pm 5\%$ under filtered, pathogen-free air, and fed ad libitum (ND, Purina lab, 38057). All mice used were male and had C57BL/6 background. *Ahr*fl/fl mice (the Jackson Laboratory, #006203) were crossed with *Alb-Cre* transgenic mice (The Jackson Laboratory) to generate *Ahr* HKO mice. *Ahr*fl/fl mice without detectable Cre genes were used as WT littermates. For chronic feeding, mice at the age of 10 to 12 weeks were fed the Lieber-DeCarli diet (Dyets Inc., #710260) with 5% (vol/vol) ethanol (36% ethanol-derived calories) or the isocaloric liquid diet ad libitum for 4 or 5 weeks. For chronic-binge feeding (NIAAA model), mice were fed the control or Lieber-DeCarli diet for 4 weeks and were additionally administered with a single dose of ethanol (5 g/kg body weight; diluted in water (50:50 vol/vol)) by gastric intubation in the early morning twice a week during the 4 weeks Lieber-DeCarli diet feeding. Pair-fed control mice were treated with tap water. The group sizes were chosen based on our experience with alcohol diet models without using a statistical method to predetermine sample size. For the experiment, mice of similar age and weights were randomly subgrouped and subjected to treatments, but the experimenters were not blinded to the genotype of the mice because of the experimental design. Validation of the line was assessed by genotyping genomic DNA from pups isolated from an ear punch of 2-week-old pups using the Maxime™ PCR PreMix (i-Taq) (iNtRON Bio, 25026). PCR primer pairs for genotyping are listed in Supplementary Table 4. Blood and liver samples taken from mice were biochemically and histo-pathologically analyzed. All animal use and studies were approved by the Institutional Animal Care and Use Committee (IACUC) of the Seoul National University (No. SNU-190515-1) and Dongguk University (No. IACUC-2022-005-1, IACUC-2022-001-1).

### Cell lines and mPHs in vitro/ex vivo
HepG2 (a human hepatocyte-derived cell line, HB-8065) and AML-12 (a mouse hepatocyte-derived cell line, CRL-2254) cells were purchased from American Type Culture Collection (ATCC) (Rockville, Maryland). HepG2 cells were maintained in the DMEM containing 10% FBS, 50

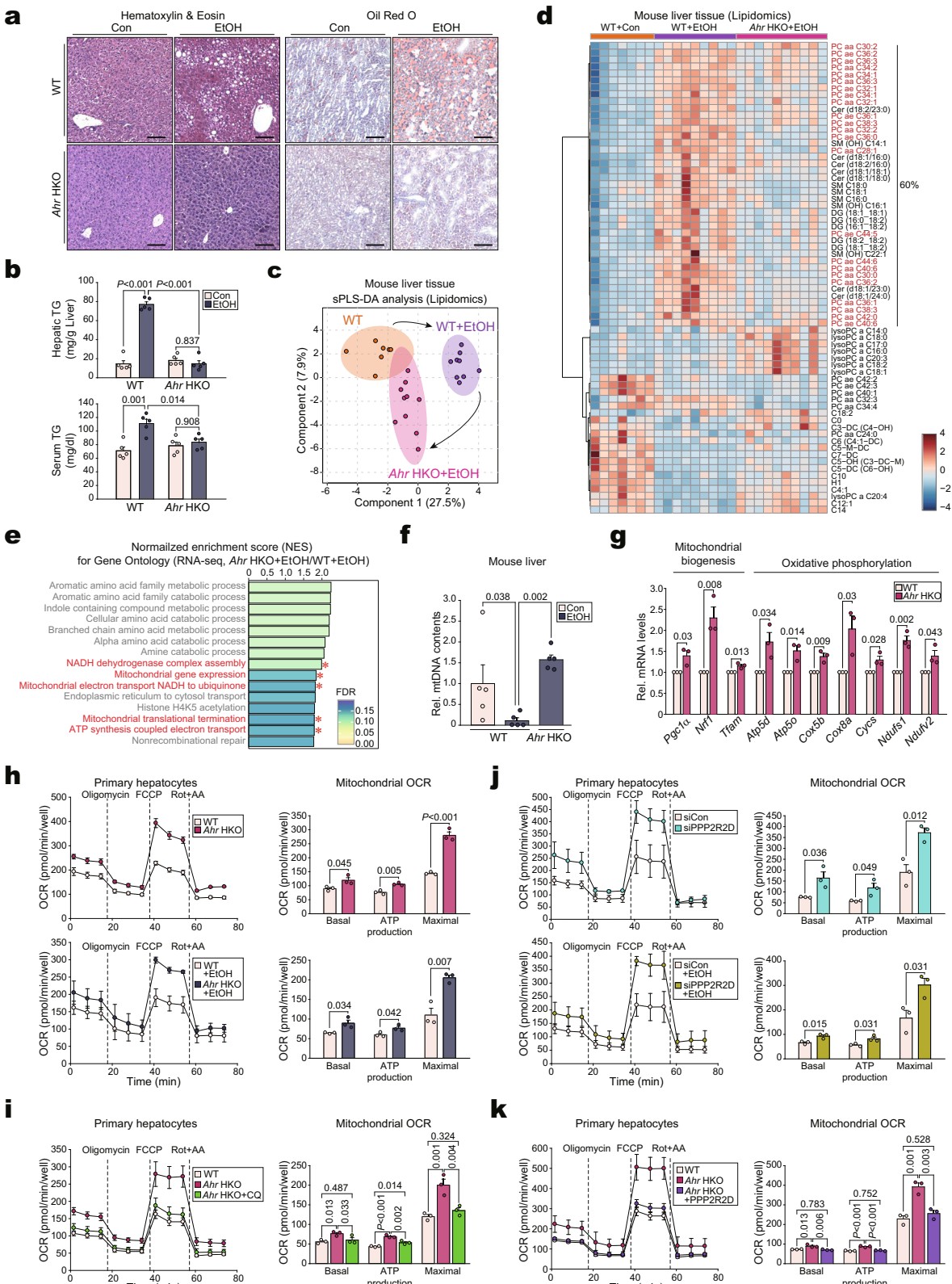

units/ml penicillin, and 50 μg/ml streptomycin. AML-12 cells were cultured in the DMEM/F-12 containing 10% FBS, insulin-transferrin-selenium X (ITSX), dexamethasone (40 ng/ml; Sigma), and the antibiotics. The cells with less than 20 passage numbers were used. mPHs were isolated from C57BL/6 mice under the guidelines of the Institutional Animal Use and Care Committee[70]. Briefly, under anesthesia with Zoletil, the liver was perfused with Ca²⁺-free Hank's buffered salt

solution (Invitrogen, Carlsbad, CA) for 10 min, followed by continuous perfusion with a 0.1% w/v collagenase (type I, Sigma). The whole liver was removed, and minced in PBS. The cell suspension was filtered through the cell strainer (70 μm) and purified with Percoll. Hepatocytes were harvested into collagen-coated plates (5 × 10⁵ cells/well) in Dulbecco's modified Eagle's medium (DMEM) containing 10% FBS, 50 units/ml penicillin, and 50 μg/ml streptomycin. Six hours afterward,

**Fig. 6 | The alteration of lipid catabolism and mitochondrial function in *Ahr* HKO and of mitochondrial respiration by hepatic ablation of either AhR or PPP2R2D. a** Representative H&E (left) or oil red O (right) staining in the liver of mice fed with Lieber-DeCarli diets for 5 weeks ($n = 5$ each). Scale bar: 100 μm. **b** Hepatic (upper) or serum (lower) TG contents in the same mice as in (**a**) ($n = 5$ each). **c** sPLS-DA score plot based on the lipidomics data using mice fed as in (**a**) ($n = 7, 9, 10$ mice). **d** Heatmap and hierarchical correlation analyses using the same data as in (**c**) ($n = 7, 9, 10$ mice; FDR <0.01; darker blue, stronger downregulation; darker red, stronger upregulation). **e** Bar graphs based on Gene Ontology analysis using the same data as in Fig. 4a ($n = 3$ each). NES and FDR are presented as bar graphs (NES >1.75, FDR <0.25). The pathways associated with mitochondrial function were highlighted in red. **f** Quantification of mtDNA content. The copy number ratio of mitochondrial/nuclear DNA was assessed by qRT-PCR assays for *mt-COXI* and *nRIP140* using total DNA samples isolated from the same mice as in (**a**) ($n = 5$ each). **g** qRT-PCR assays for transcripts associated with mitochondrial function in mPHs isolated from WT or *Ahr* HKO mice ($n = 3$ each). **h**–**k** OCRs in mPHs. The OCRs of basal, ATP production (via oligomycin addition), maximal (with FCCP), and non-mitochondrial respiration (with rotenone + antimycin A [Rot + AA]) were determined using an XFp Extracellular Flux Analyzer. Real-time triplicate readings (left) and calculated mitochondrial respiration rates (right) are shown ($n = 3$ each). mPHs were treated with control (upper) or 100 mM ethanol (lower) for 24 h (**h**); mPHs were treated with 10 μM CQ for 24 h (**i**); the mPHs were either untreated (control) (upper) or treated with 100 mM ethanol (lower) for 24 h after transfection with siPPP2R2D (or siCon) for 24 h (**j**); and mPHs were transfected with PPP2R2D plasmids for 24 h (**k**). The OCRs were normalized to the cell counts. Values are expressed as means ± SEM. Significantly different compared to Con or WT. Data were analyzed via one-way ANOVA with Tukey HSD (**b**, **k**), one-way ANOVA with LSD (**f**, **i**), or two-tailed Student's *t*-test (**g**, **h**, **j**). Source data are provided as a Source Data file.

mPHs are ready for experimental use. For ex vivo experiments, mPHs were isolated from mice fed either a control diet or a Lieber-DeCarli alcohol liquid diet for 5 weeks. Hepatocyte-derived cells and primary hepatocytes were treated with 100 mM ethanol for 48 h (freshly diluted in media), 100 μM kynurenine for 12 h, and 100 nM FICZ for 4 h or for indicated times. To prevent ethanol evaporation during exposure, each culture dish was tightly wrapped with Parafilm.

### Isolation of HSCs and Kupffer cells

Primary HSCs were isolated from male C57BL/6 mice. Briefly, livers were perfused in situ with Ca²⁺-free Hank's balanced saline solution at 37 °C for 15 min and then perfused with the solution containing 0.05% collagenase and Ca²⁺ for 15 min at a flow rate of 10 mL/min. Perfused livers were minced, filtered through a 70-μm cell strainer (BD Biosciences, Franklin Lakes, NJ), and centrifuged at $50 \times g$ for 5 min to separate hepatocytes. The supernatant was centrifuged further at $500 \times g$ for 10 min, resuspended in Ficoll plus Percoll (1:10; GE Healthcare), and centrifuged at $1400 \times g$ for 17 min. HSCs were collected from the interface. The cells were cultured in Dulbecco's modified Eagle medium containing 10% fetal bovine serum, 100 U/mL penicillin, and 100 mg/mL streptomycin at 37 °C in a humidified atmosphere containing 5% CO₂. Kupffer cells were isolated using Optiprep[71]. The nonparenchymal cells are resuspended in 20% Optiprep. HBSS and 11.5% Optiprep were layered on the cell suspension and centrifuged at $1811 \times g$ for 17 min. Kupffer cell fraction was obtained from the layer between 20% Optiprep and 11.5% Optiprep.

### RNA-seq analysis

For RNA-seq analysis, liver from the mice of litter and *Ahr* HKO fed either a control diet or Lieber-DeCarli alcohol liquid diet for 5 weeks was used ($n = 7, 9, 10$ mice; three samples randomly selected from each group for RNA-seq analysis, whereas all samples were used for metabolomics analysis). Total RNA concentration was calculated by Quant-IT RiboGreen (Thermo Fisher Scientific, R11490). To assess the integrity of the total RNA, samples were run on the TapeStation RNA screen tape (Agilent Technologies, 5067-5576). Only high-quality RNA preparations, with RIN >7.0, were used for RNA library construction. A library was independently prepared with 1 μg of total RNA for each sample by TruSeq Stranded mRNA LT Sample Prep Kit. The first step in the workflow involved purifying the poly-A-containing mRNA molecules using poly-T-attached magnetic beads. Following purification, the mRNA was fragmented into small pieces using divalent cations under elevated temperatures. The cleaved RNA fragments were copied into first-strand cDNA using SuperScript II reverse transcriptase (Invitrogen, 18064014) and random primers. This was followed by second-strand cDNA synthesis using DNA Polymerase I, RNase H and dUTP. These cDNA fragments then went through an end repair process, the addition of a single "A" base, and then ligation of the adapters. The products were then purified and enriched with PCR to create the final cDNA library. The libraries were quantified using KAPA Library Quantification Kits for Illumina Sequencing platforms according to the qRT-PCR Quantification Protocol Guide (Kapa Biosystems, KK4854) and qualified using the TapeStation D1000 ScreenTape (Agilent Technologies, 5067-5582). Indexed libraries were submitted to an Illumina NovaSeq 6000 (Illumina, Inc.), and the paired-end ($2 \times 100$ bp) sequencing was done by Macrogen Incorporated. For RNA-seq analysis, the raw "fragments per kilobase million (FPKM)" values were processed and normalized by logarithm and quantile normalization method. DEGs were selected as the genes with adjusted *P* value <0.05 or 0.25, or *P* value <0.05 with FC >1.5 or 2. False discovery rate (FDR) was controlled by adjusted *P* value and *P* value was derived from modified fisher's exact test. Heatmap and hierarchical correlation analysis, PCA, hierarchical clustering with Pearson's correlation, and volcano plot of the DEGs were analyzed using R software 3.6.2. Statistically enriched signaling pathways of clustered DEGs were ranked and categorized according to the KEGG pathway using the KEGG database (http://www.genome.jp/kegg/). "WikiPathways", "KEGG", "REACTOME", "Gene Ontology", and "NADLER" from Molecular Signature Database (MSigDB v7.4, http://software.broadinstitute.org/gsea/msigdb) and GSEA leading-edge analysis was employed using GSEA 4.1.0 software with the "Signal2Noise" metric to generate a ranked list and a "gene set" permutation type. FDR was used for statistical significance assessment of normalized enrichment score (NES). The criterion for statistical significance was set at FDR <0.25. The heatmap represents the respective leading-edge subsets of the most upregulated genes. The raw transcriptome data have been deposited in the Gene Expression Omnibus (GEO) under the accession code GSE179398.

### GSEA

Transcriptome data from mice fed on a Lieber-DeCarli alcohol liquid diet for 4 weeks from GEO (https://www.ncbi.nlm.nih.gov/geo/) (GSE40334) were analyzed using GSEA 4.1.0 software. "KEGG" from MSigDB (http://software.broadinstitute.org/gsea/msigdb) v7.4 and GSEA leading-edge analysis was employed using GSEA 4.1.0 software. FDR was used for the statistical significance assessment of NES. Gene sets with FDR <0.25 were considered statistically significant.

### Metabolomics and lipidomics analysis

Targeted metabolomics of liver or serum samples from the mice of litter and *Ahr* HKO fed either a control diet or Lieber-DeCarli alcohol liquid diet for 5 weeks ($n = 7, 9, 10$ mice) was performed using tandem mass spectrometry with the Biocrates MxP® Quant 500 Kit (Biocrates, Innsbruck, Austria). The MxP® Quant 500 kit has been developed to quantify 630 endogenous metabolites and lipids belonging to 26 biochemical classes. The MxP Quant 500 kit combines liquid chromatography-tandem mass spectrometry (LC-MS/MS) of 13 compound classes, basically, amino acids, amino acid derivatives, small

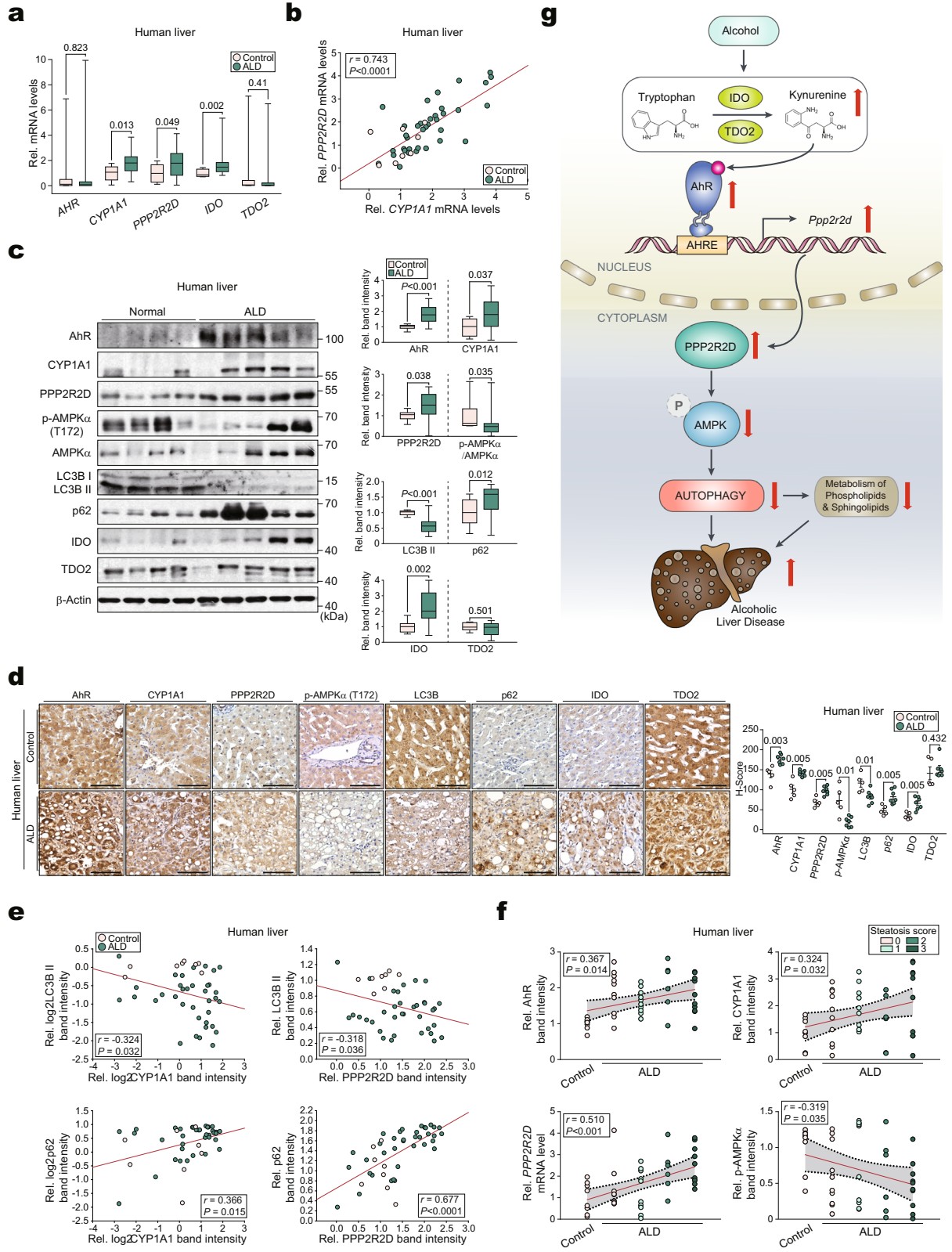

molecules, and free fatty acids, followed by flow injection analysis tandem mass spectrometry (FIA-MS/MS) of 12 lipid classes and hexoses, using a 5500 QTRAP® instrument. Briefly, a 96-well-based sample preparation device was used to quantitatively analyze the metabolite profile in the samples. This device consists of inserts that have been spotted with internal standards, and a predefined sample amount was added to the inserts. Next, a phenylisothiocyanate solution was added

to derivatize some of the analytes (e.g., amino acids), and after the derivatization was completed, the target analytes were extracted with an organic solvent, followed by a dilution step. The obtained extracts were then analyzed by FIA-MS/MS and LC-MS/MS methods using multiple reaction monitoring (MRM) to detect the analytes. Data were quantified using appropriate MS software (version window 10, Sciex Analyst®) and imported into Biocrates MetIDQ™ software (version

**Fig. 7 | Analyses of identified AhR targets in the livers of ALD patients. a** qRT-PCR assays for *AHR* or other identified targets in the liver biopsy samples from ALD patients (control: $n = 8$, ALD: $n = 36$). Data values in box-and-whisker plots indicate median (mid-line), boxes indicating the Q1 and Q3 ranges, and the whiskers as minimum and maximum values. **b** Correlations between *CYP1A1* and *PPP2R2D* using the same data as in (**a**). Each point represents one sample (control: $n = 8$, ALD: $n = 36$). **c** Immunoblots of the newly-identified AhR targets or autophagy markers using the same human samples as in (**a**) (left) and their quantifications (right) (control: $n = 8$, ALD: $n = 36$). Data values in box-and-whisker plots indicate median (mid-line), boxes indicating the Q1 and Q3 ranges, and the whiskers as minimum and maximum values. **d** Representative immunohistochemical images of the newly-identified targets or autophagy markers in the liver specimens from ALD patients (*left*) (control: $n = 5$, ALD: $n = 7$) and their H-scores (right). Scale bar: 100 μm. **e** Correlations between CYP1A1 (or PPP2R2D) and autophagy markers using the same data as in (**c**). Each point represents one sample (control: $n = 8$, ALD: $n = 36$). **f** Linear regression analysis between histologic steatosis score (0–3) and expression level of AhR (RMSE = 0.544), CYP1A1 (RMSE = 0.981), *PPP2R2D* (RMSE = 0.932), or p-AMPKα (RMSE = 0.448) using the same data as in (**a**) or (**c**). The red line is the regression line, and the gray area between the black dotted lines indicates the 95% confidence intervals of the fit. Each point represents one sample (control: $n = 8$, ALD: $n = 36$). **g** Proposed scheme illustrating the mechanism by which alcohol overexpression of AhR dysregulates autophagy in hepatocytes. Values are expressed as means ± SEM. Significantly different compared to Control. Data were analyzed by two-tailed Student's *t*-test (part of both **a** and **c**), two-tailed Mann–Whitney test (part of both **a** and **c** and **d**), two-tailed Pearson correlation (**b**, part of **e**, and **f**), or two-tailed Spearman correlation (part of **e**). Source data are provided as a Source Data file.

Oxygen; Biocrates Life Sciences AG, Innsbruck, Austria) for calculating analyte concentrations, data assessment, and compilation. Metabolic analyses were done using the Metaboanalyst function Metabolite Set Enrichment Analysis (MSEA, version 4.0) and R software 3.6.2. sPLS-DA, heatmap, and Pearson's correlation was performed using R software 3.6.2. Metabolites with adjusted P values less than 0.05 or 0.01 were selected. FDR was controlled by the adjusted *P* value. Raw data are provided in Supplementary Data 1.

## Materials

Antibodies to p-Thr172-AMPKα (2535, RRID:AB_331250, dilution 1:10000, clone: 40H9), AMPKα (2532, RRID:AB_330331, dilution 1:10000), LKB1 (3047, RRID: AB_2198327, dilution 1:10000, clone: D60C5), p-Ser428-LKB1 (3482, RRID: AB_2198321, dilution 1:10000, clone: C67A3), ULK1 (8054, RRID: AB_11178668, dilution 1:10000, clone: D8H5), p-Ser555-ULK1 (5869, RRID: AB_10707365, dilution 1:10000, clone: D1H4), and Lamin A/C (2032, RRID:AB_2136278, dilution 1:10000) were from Cell Signaling (Danvers, MA, USA). Anti-CYP1A1 (13241-1-AP, RRID: AB_2877928, dilution 1:5000), anti-IDO (13268-1-AP, RRID: AB_2123444, dilution 1:5000), and anti-TDO2 (15880-1-AP, RRID: AB_2827610, dilution 1:5000) were from Proteintech (Rosemont, IL, USA). Antibodies directed against IDO (MAB5412, RRID: AB_2123547, dilution 1:5000, clone: 10.1), LC3B (L7543, RRID: AB_796155, dilution 1:1000), β-Actin (A5441, RRID: AB_476744, dilution 1:10000, clone: AC15), ʟ-kynurenine (K8625), chloroquine (CQ, C6628), 1-methyl-ʟ-tryptophan (1-MT, 447439), 680C91 (SML0287), and 5-aminoimidazole-4-carboxamide 1-β-ᴅ-ribofuranoside (AICAR, A9978) were supplied from Sigma (St. Louis, MO, USA). Anti-AhR antibodies (MA1-514, RRID: AB_2273723, dilution 1:5000, clone: RPT1; MA1-513, RRID: AB_2223958, dilution 1:5000, clone: RPT9) were purchased from Invitrogen (Carlsbad, CA, USA). Anti-PPP2R2D (GTX116609, RRID: AB_10615217, dilution 1:5000, clone: N2C3) was supplied from GeneTax (Irvine, CA, USA), whereas anti-LC3B (NB100-2220, RRID: AB_10003146, dilution 1:1000) antibody was from Novus Biologicals (Littleton, CO, USA). Antibody against p62 (H00008878-M01, RRID: AB_437085, dilution 1:10000, clone: 2C11) was purchased from Abnova (TPE, Taiwan). Antibody to normal mouse IgG (sc-2025, RRID: AB_737182, dilution 1:1000) was supplied by Santa Cruz Biotechnology (Santa Cruz, CA, USA). 6-Formylindolo 3, 2-b carbazole (FICZ, BML-GR206-0100) was supplied from Enzo Life Science (Farmingdale, NY, USA). As secondary antibodies, horseradish peroxidase-conjugated goat anti-rabbit IgGs (G21234, RRID: AB_1500696, dilution 1:10000) and goat anti-mouse IgGs (G21040, RRID: AB_2536527, dilution 1:10000) were purchased from Invitrogen (Carlsbad, CA).

## Histological analysis

**Hematoxylin & eosin staining and immunohistochemistry.** Mouse liver tissues were fixed in 10% formalin, embedded in paraffin, cut into 4 μm thick sections, and were mounted on slides. Tissue

sections were stained with Hematoxylin & Eosin for morphology analysis. Tissue sections were immunostained with antibodies directed against AhR, CYP1A1, LC3B, p62, IDO, TDO2, p-Thr172AMPKα, and PPP2R2D. Mouse liver samples were fixed and immediately frozen with liquid nitrogen-cooled isopentane after dissection and were sectioned on a cryostat microtome for immunofluorescence. Immunofluorescence staining was done using antibodies specific to AhR and CYP1A1 with DAPI. Tissues were imaged with an Eclipse Ti-U inverted microscope equipped with a DS-F1i digital microscope camera in conjugation with NIS-Elements F software (version 4.0, Nikon, Tokyo, Japan).

**Oil Red O staining.** Cryosections of snap-frozen liver were stained with an Oil Red O stain kit (ScyTek, ORK-1-IFU) for neutral TG and lipid analyses.

## Transmission electron microscopy (TEM)

The liver was fixed in Karnovsky's fixative immediately after isolation. Samples were then post-fixed by 1% osmium tetraoxide in 0.05 M sodium cacodylate buffer. After overnight incubation with 0.5% uranyl acetate, the samples were dehydrated in ethanol and propylene oxide and polymerized in Spurr's resin at 70 °C. The observation was performed by the National Instrumentation Center for Environmental Management at Seoul National University. The images were taken using a JEM-1010 transmission electron microscope (JEOL, Japan).

## Blood chemistry

ALT activity, AST activity, and TGs levels were analyzed using Spectrum, an automatic blood chemistry analyzer (Abbott Laboratories, Abbott Park, IL).

## Hepatic TGs measurement

A fraction of mouse liver (0.5 g) was homogenized in 0.1 M Tris-acetate buffer (pH 7.4) containing 0.1 M potassium chloride and 1 mM EDTA. Six volumes of chloroform/methanol (2:1) were added. After vigorous stirring, the mixtures were incubated on ice for 1 h and then centrifuged at 800 × *g* for 3 min. The resulting lower phase was aspirated. The TG content was determined using Sigma Diagnostic Triglyceride Reagents (Sigma)[72]. In the NIAAA mice model, TG levels were analyzed with a Triglyceride Quantification Kit (Abcam, ab65336) according to the manufacturer's instructions.

## RNA isolation and qRT-PCR assays

Total RNA was extracted using TRIzol (Invitrogen, Carlsbad, CA, USA), and was reverse transcribed[73]. qRT-PCR was carried out using ABI Step One Plus Real-Time PCR System and 48-well optical reaction plates (Applied Biosystems, Foster City, CA, USA). Gapdh, 18 S RNA, and β-Actin were used as a normalized control. Gene expression was assessed by qRT-PCR analysis using the primers shown in Supplementary Table 5.

### ChIP assay

AML-12 cells were exposed to 100 mM ethanol for 24 h, and then formaldehyde was added to the cells to a final concentration of 1% for cross-linking of chromatin. The ChIP assay was performed according to the EZ-ChIP assay kit protocol (Upstate Biotechnology, Lake Placid, NY, USA). qRT-PCR was done using the primers flanking putative AhR binding sites located in the promoter region of the mouse *Ppp2r2d* gene. IgG immunoprecipitation represents negative control. The primer sequences used for the qRT-PCR analysis are listed in Supplementary Table 5.

### Flow cytometry

HepG2 cells were exposed to 300 μM oleic acid for 24 h after transfection with AhR plasmid for 48 h, and the cells were harvested by trypsinization. After washing with PBS, the cells were stained with 30 nM Nile Red or 0.05 μg/ml Rhodamine 123 for 30 min in cell culture media containing 20% FBS. In each analysis, 20,000 gated events were recorded. The fluorescence intensity in the cells was assessed using the FACS Calibur II flow cytometer and the CellQuest™ Pro software (version 6.0, BD Biosciences, San Jose, CA, USA). A schematic representation of the gating strategy can be found in Supplementary Fig. 9.

### Autophagic flux assays using adenoviral construct

Mouse primary hepatocytes were infected with adenovirus encoding a red fluorescent protein-green fluorescent protein-microtubule-associated protein light chain 3 (Ad-mcherry-GFP-LC3) in DMEM containing 10% FBS for 12 h. After the experiments, the cells were fixed with paraformaldehyde for 30 min, followed by samples being collagen-coated coverslips slipped with a mounting medium. After incubation, the samples were examined using a laser-scanning confocal microscope (Nikon, Eclipse Ti-U/ Yokogawa, CSU-X1, Japan).

### Immunoblot analysis

Cells were centrifuged at $3000 \times g$ for 3 min and allowed to swell after the addition of lysis buffer in ice for 30 min. The lysates were centrifuged at $10,000 \times g$ for 10 min to obtain supernatants. Proteins were separated by 7.5%, or 15% SDS-polyacrylamide gel electrophoresis and were transferred onto nitrocellulose membranes (Millipore, Bedford, MA). The membrane was blocked with 5% non-fat dried milk in Tris-buffered saline and Tween 20 (TBST) (20 mM Tris-HCl, 150 mM NaCl, and 0.1% Tween 20, pH 7.5) for 1 h, and incubated overnight with primary antibodies at 4 °C. After washing with TBST buffer, membranes were incubated with a horseradish peroxidase-conjugated anti-rabbit or -mouse IgG secondary antibody for 1 h at room temperature. The protein bands were visualized using an enhanced chemiluminescence system (GE Healthcare, Buckinghamshire, UK). Equal loading of samples was verified by immunoblotting for β-Actin or Lamin A/C. Quantifications were done by scanning densitometry of the immunoblots and β-Actin or Lamin A/C normalization.

### Cellular respiration assays

OCRs were measured in mPHs according to the manufacturer's recommended protocol of Seahorse XFp Extracellular Flux Analyzer (SeahorseBioscience, MA). The cells were harvested into collagen-coated plates ($2 \times 10^4$ cells/well) in DMEM containing 10% FBS, 50 units/ml penicillin, and 50 μg/ml streptomycin. One hour before the experiment, cells were washed with an XF assay base medium and placed into a non-CO₂ incubator maintained at 37 °C before completion of probe cartridge calibration. During the time course of the experiment, the oxygen concentration was measured overtime periods of 3 min at 3-min intervals. The rates of basal, uncoupled (by addition of 2 μM oligomycin), maximal (with 2 μM FCCP, carbonyl cyanide *p*-trifluoromethoxyphenylhydrazone), and non-mitochondrial respiration (with rotenone plus antimycin A, 0.5 μM each) were measured using an XFp Extracellular Flux Analyzer.

### Genomic DNA extraction

Genomic DNA was extracted using AccuPrep® Genomic DNA Extraction Kit (BIONEER, Daejeon, Korea) according to the manufacturer's recommended protocol. Both $Ahr^{fl/fl}$-unexcised and $Ahr^{fl/fl}$-excised alleles were examined by PCR using the primers shown in Supplementary Table 4 and mitochondrial DNA content was measured by qRT-PCR using the primers shown in Supplementary Table 5.

### Preparation of nuclear extracts

Cells were allowed to swell after the addition of hypotonic buffer containing 10 mM HEPES (pH 7.9), 10 mM KCl, 0.1 mM EDTA, 0.5% NP-40, 1 mM DTT, and 0.5 mM PMSF. The lysates were incubated for 10 min on ice and centrifuged at $7200 \times g$ for 5 min. Pellets containing crude nuclei were then resuspended in an extraction buffer containing 20 mM HEPES (pH 7.9), 400 mM NaCl, 1 mM EDTA, 10 mM DTT, and 1 mM PMSF fluoride, and were incubated for 30 min on ice. The samples were centrifuged at $15,800 \times g$ for 10 min to obtain supernatants.

### Transfection of plasmids and siRNAs

FLAG-AhR and -PPP2R2D expression clones were supplied from GeneCopoeia (Rockville, MD, USA). The empty plasmid, pcDNA3.1, was used for mock transfection. For gene knockdown assays, scrambled control siRNA, and siRNAs specifically directed against AhR (L-044066-00-0005) and PPP2R2D (L-057039-02-0005) were purchased from Dharmacon (Lafayette, CO, USA). The cells were transfected with plasmid and/or siRNA using Lipofectamine 2000 transfection reagent (Invitrogen, Carlsbad, CA, USA) or FuGENE® HD Reagent (Promega, Madison, WI, USA) in accordance with the manufacturer's procedure.

### Reporter gene assays

The upstream promoter region of the mouse *Ppp2r2d* gene containing up to −2.0 kb was cloned into the pGL3 luciferase vector. A mutation of AHRE in the mouse gene was done by replacing the sequence of putative AhR binding element (5′-TGCGTG-3′) located between −1154 and −1149 bp (Mut1); or between −1064 and −1059 bp (Mut2), respectively. AML-12 cells were transfected with pGL3-*Ppp2r2d* for 24 h with either siCon or siAhR, and luciferase activity was measured by adding a luciferase assay reagent (Promega, Madison, WI).

### Statistical analysis

Values are expressed as means ± standard error of the mean (SEM). Statistical significance was tested using a two-tailed Student's *t*-test, Mann–Whitney test, or one-way ANOVA with Tukey's Honestly Significant Difference (Tukey's HSD) or least significant difference (LSD) multiple comparisons (more than two groups) where appropriate. Coefficients of correlation (r) were determined by the Pearson correlation or Spearman correlation method. Differences were considered significant at $P < 0.05$. Statistical analyses were performed using IBM SPSS Statistics 25.0 software or Prism (version 7.0, GraphPad Software). No data were excluded when conducting the final statistical analysis.

### Reporting summary

Further information on research design is available in the Nature Research Reporting Summary linked to this article.

## Data availability

The raw transcriptome generated in this study have been deposited in the NCBI's GEO database under accession code GSE179398. The public transcriptome data used in this study are available in the NCBI's GEO database under accession code GSE40334[74]. MSigDB v7.4: Gene sets were obtained from MSigDB v7.4 for "WikiPathways", "KEGG", "REACTOME", "Gene Ontology", and "NADLER"(http://software.broadinstitute.org/gsea/msigdb). The canonical motifs of AhR binding sites (AHRE) were defined from the Jaspar 2018 database (http://jaspar.genereg.net). Enriched KEGG pathways were annotated using

the KEGG database (http://www.genome.jp/kegg/). Metabolomics and lipidomics data are provided in Supplementary Data 1. All data supporting the findings of this study are available within the article and its Supplementary Information files. Any other data, mouse lines, and materials generated or used in this study are available from the corresponding author upon reasonable request. Source data are provided with this paper.

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

## Acknowledgements

This research was supported by the National Research Foundation (NRF) grant funded by the Korea government (NRF-2021R1A2B5B03086265 to S.G.K.; NRF-2021R1A6A3A01086434 to Y.S.K.; NRF-2021R1A2C2005820 and NRF-2021M3A9E4021818 to W.K.).

## Author contributions

Y.S.K. designed and performed experiments, analyzed and interpreted data, and drafted the manuscript. B.K. performed experiments and analyzed and interpreted data. D.J.K. and J.-Y.C. performed lipidomics, metabolomics analysis, and interpretation of data. J.T. performed experiments. C.Y.H. designed experiments and interpreted data. W.K. collected, analyzed, and provided human samples. S.G.K.: designed research, supervised experiments, drafted and critically reviewed the manuscript for important intellectual content, obtained funding,

administrative, technical, or material support, and study supervision. All authors read and approved the final manuscript.

## Competing interests

The authors declare no competing interests.
