## [Peer Review File · Nature Communications]

Title: Induction of the hepatic aryl hydrocarbon receptor by alcohol dysregulates autophagy and phospholipid metabolism via PPP2R2DREVIEWER COMMENTS

Reviewer #1 (Remarks to the Author):

The manuscript "AhR induction by alcohol dysregulates autophagy and phospholipids metabolism via PPP2R2D" by Yum Seok Kim et al, demonstrated that alcohol consumption increased endogenous AhR ligand kynurenine level, resulting in inhibition of autophagy in hepatocytes. Mechanistically, AhR over-induction by alcohol inhibited autophagy in hepatocytes through AMPK α , as mediated by Ppp2r2d gene transactivation, leading to the change of phospho/sphingo-lipid metabolism, which is a novel concept. However, there are several significant issues the authors need to address.

Comments

1. The relevance of direct EtOH exposure in primary hepatocytes or hepatocyte cell line such as those described in Fig.1i, 1j, 2g, Extended Data Fig. 3a, Fig. 5c is highly questionable. Direct EtOH exposure is not toxic to cultured primary hepatocytes.
2. Data in Extended data Fig. 2g should be repeated in the context of ALD.
3. In Fig. 3, was the liver and serum level of Kynurenine increased in hepatocyte Ahr KO mice upon EtOH exposure?
4. The effect of AhR on autophagy was an association, the authors did not provide sufficient data to show the functional relevance of autophagy in phenotypic exhibition.
5. AhR is differentially expressed in different liver cell types. It is necessary to provide data showing AhR expression in different liver cell types (hepatocytes, hepatic stellate cells, Kupffer cells, and sinusoidal endothelial cells) upon alcohol challenge. Does AhR in other liver cells play a role in ALD?
6. The legends of Extended Data Figure 2 a-d are incorrect.
7. It is necessary to provide mRNA or protein level of AhR in hepatocytes from HEP-AhR knockout mice vs. flox controls to verify the AhR knockout efficiency in hepatocytes.
8. What is the effect of kynurenine in liver autophagy in mice fed on Lieber-DeCarli diet?
9. What is the effect of IDO/TDO2 knockdown/inhibition on hepatocytes isolated from Lieber-DeCarli diet-treated mice?
10. Please explain why AHR mRNA expression was not changed between ALD and Normal liver (Figure 7a), but the protein expression was increased (Figure 7c).

Reviewer #2 (Remarks to the Author):

Authors investigated the role and potential mechanisms of aryl hydrocarbon receptor (AhR) in regulating AMPK and autophagy in the early stage of alcoholic liver disease (ALD). They utilized a Lieber-DeCarli alcohol diet feeding mouse model that can mimic early stage of steatosis of ALD as well as in vitro cell culture (primary hepatocytes and AML12) and correlative studies of human ALD liver samples. Unbiased transcriptomic and metabolomics studies were used. They identified increased levels of kynurenine, an endogenous AhR ligand in ALD mouse models and ethanol-treated hepatocytes. They further identified Ppp2r2d, a regulatory subunit of PP2A, as a transcriptional target of AhR, which was upregulated in alcohol fed mouse livers resulting in decreased phosphorylation of AMPK and possible decreased hepatic autophagy. Liver-specific AhR knockout mice fed with alcohol diet also had decreased hepatic steatosis and liver injury. The authors concluded that kynurenine-mediated AhR activation by alcohol may inhibit autophagy flux in hepatocytes through Ppp2r2d-mediated AMPK α dephosphorylation. Overall, the identification of AhR- Ppp2r2d-AMPK α axis in regulating autophagy in ALD is novel and interesting. Most data were of good quality and comprehensive/complementary to support the role of AhR in the early ALD. However, several limitations/weaknesses were noticed for this study as the mouse model only can reflect the early stage but not the more severe stage of ALD, and the later (such as alcoholic hepatitis) is more important in clinic. Some of the data were also questionable and need to be further improved/clarified.

Specific comments:

1. Figure 1G, better quality IHC staining images should be provided for AHR (hardly to see any nuclear AHR staining). It seems that CYP1A1 staining showed a clear zonation pattern. Was CYP1A1 staining enriched in the central vein area? Ideally authors should show the same liver areas for the IHC staining of both AHR and CYP1A1 to show the correlation.
2. Supplemental Figure 2, in addition to the genotyping PCR data, western blot analysis data for liver AHR should be provided (even if some truncated proteins may be present as authors claimed). Supplemental Figure 2E, the statement "Alcohol exposure inhibited the autophagy markers" was too vague. Authors should perform densitometry analysis and clearly stated the changes on LC3B-II and p62 after alcohol feeding.
3. Figure 2C, the quality of the EM image was poor and hardly to see any autophagy-related structures. Why the control Ahr HKO mice had a lot of LDs, which was not consistent with Figure 6b? Was the image representative as Ahr HKO mice. A better rigorous controlled autophagic flux assay was lacking for 2D, F, G, as without a lysosomal inhibitor these data were difficult to interpret. As it would be difficult to perform this in vivo (requires another feeding), authors should consider at least to repeat the autophagic flux assay in vitro primary hepatocytes and AHR knockdown AML12 cells with ethanol treatment and with/without chloroquine.
4. The levels of p62 in Ahr KO hepatocytes and KD AML-12 cells was almost undetectable and very striking (Figure 2G and Figure 3I). As it is well known p62/SQSTM1 is also subjected to transcription regulation, it will be helpful to provide the mRNA data of p62/SQSTM1 given AHR is a transcription factor. On the same line, it will be interesting to see whether other autophagy-related genes would be changed at the transcription levels after manipulating AHR such as providing a heatmap for autophagy-

related genes from the RNAseq analysis? Figure 3H, p62 levels should be provided to complete the experiments.

5. Figure 6, the improved mitochondria biogenesis in Ahr HKO mice was interesting. It was known that TFEB regulated PGC1a and biogenesis of mitochondria and lysosome, which is impaired by alcohol. This should be discussed or explored. Autophagy is a dynamic process and autophagic flux can be influenced by the early autophagosome synthesis or late stage lysosomal functions/biogenesis, and it has been reported that alcohol consumption may affect the later state autophagy too (PMID:29782848). This should be discussed to clearly reflect the progress of this current study to the ALD field. AMPK regulates autophagy by inhibiting mTORC1 activity or ULK1 phosphorylation? These downstream events after alcohol should be explored to provide additional mechanistic insights.

6. For the human ALD samples, more detail description on these ALD samples should be provided as ALD ranges from early steatosis to later stage fibrosis, inflammation and alcoholic hepatitis. This is particular important, as the "ALD mouse model" by Lieber-DeCarli diet feeding only represent/recapitulate the early steatosis phenotype of ALD (mild or no inflammation and fibrosis). This is a limitation of this study. As the human ALD samples seem to be from late stage ALD based on the sole H&E staining imaging data (Supplemental figure 7A). The impact of this manuscript may be improved if author can apply other relative more severe ALD mouse models such as the chronic plus binge NIAAA model.

7. Another limitation for this study was that authors should be aware that both AHR and AMPK have multiple targets/pathways rather than regulating autophagy, so other pathway may be involved for the ALD phenotypes of AHR KO mice that they observed in addition to autophagy. This should be discussed.

8. Authors should do a better job to summarize/discuss on recently findings on Autophagy and AHR in ALD (some key papers were even not mentioned; PMID: 3408211, 34454169; 29782848).

Reviewer #3 (Remarks to the Author):

Kim et al. have demonstrated the role of AhR in regulation of lipid metabolism in alcohol-fed mice. The metabotropic AhR overinduction by alcohol inhibits autophagy in hepatocytes through AMPK α , as mediated by Ppp2r2d gene transactivation, revealing a new AhR-dependent metabolism of phospho/sphingo-lipids. To draw a more solid conclusion, several concerns need to be addressed.

1. Metabolic profiling of serum, liver (liver biopsy tissue) and feces from clinical cohort are necessary to validate the increased levels of kynurenine in ALD patients. Previous studies have proved that high productions of bacterial tryptophan metabolites were associated with an improvement of hepatic injury, which are not consistent with this study (e.g., PMID: 33004548).

2. Previous studies established the role of gut microbiota-metabolites in activation of intestinal AHR for the alleviation of alcohol-induced liver injury (PMID: 34454169; 33004548), which was not in accordance

with the hepatic outcomings. Please explain the discrepancy between the effects of AHR on liver and intestine or whether there is any crosstalk between the hepatic and intestinal AHR signaling.

3. It is observed that hepatic expression of IDO was elevated in mice intervened with alcohol, thus, please provide some mechanistic links between the alcohol and IDO or TDO2.

4. As shown in Figure 3a, the “tyrosine metabolism” was significantly upregulated in ethanol-treated group compared with control, I just wonder whether “tyrosine metabolism”-related metabolites changed accordingly?

5. Please provide the immunoblotting results of p62 in Figure 3h.

6. Please provide the liver weights of WT and Ahr HKO groups other than the liver/body weight ratio.

7. The tissue microarray putting all the human tissue samples into one array is recommended to see the immunohistochemical differences of the molecular targets between the two groups more clearly.

8. The length of scale bars throughout the manuscript are not consistent although they all represent 100 μm .

9. Seven mice in control group and 9 in alcohol-treated were included for the metabolomics analysis, why only three in each group were chosen for the RNA-seq analysis.

10. There are mistakes in the figure legends in Extended Data Figure 2, 4.

Response to Referees Letter

Reviewer #1 (Remarks to the Author):

The manuscript “AhR induction by alcohol dysregulates autophagy and phospholipids metabolism via PPP2R2D” by Yum Seok Kim et al, demonstrated that alcohol consumption increased endogenous AhR ligand kynurenine level, resulting in inhibition of autophagy in hepatocytes. Mechanistically, AhR over-induction by alcohol inhibited autophagy in hepatocytes through AMPK α , as mediated by Ppp2r2d gene transactivation, leading to the change of phospho/sphingo-lipid metabolism, which is a novel concept. However, there are several significant issues the authors need to address.

Comments

1. The relevance of direct EtOH exposure in primary hepatocytes or hepatocyte cell line such as those described in Fig.1i, 1j, 2g, Extended Data Fig. 3a, Fig. 5c is highly questionable. Direct EtOH exposure is not toxic to cultured primary hepatocytes.

Answer: In vitro alcohol exposure models are well-established methods; the authors published a series of papers using these methods. As an example, we previously reported hepatocytic cell death in alcoholic liver disease using primary hepatocytes (Heo, M. J. *et al.*, *Gut*, 2019), and a number of other studies have also used primary hepatocytes to verify ethanol-induced hepatic injury such as apoptosis, inflammation, altered lipid metabolism, and oxidative stress (Ding, W.X. *et al.*, *Gastroenterology*, 2010; Nourissat, P. *et al.*, *Hepatology*, 2008; Schulze, R. J. *et al.*, *Hepatol. Commun.*, 2017; Lee, Y. J. *et al.*, *PLoS One*, 2019). Usually, the data using primary hepatocytes are in agreement with findings in hepatocyte cell lines, ex vivo primary hepatocytes, and liver tissues. In the present study, we also confirmed this. Therefore, the research outcomes are clearly acceptable.

2. Data in Extended data Fig. 2g should be repeated in the context of ALD.

Answer: The authors appreciate the reviewer’s helpful comment. As the reviewer suggested, the effect of AhR under ethanol conditions on autophagy regulation was analyzed, and the data was added in the revised Supplementary Fig. 2k. As expected, the accumulation of LC3-II in the cells with AhR ablation was significantly higher than in control cells, indicating that AhR lessened autophagy flux.

3. In Fig. 3, was the liver and serum level of Kynurenine increased in hepatocyte Ahr KO mice upon EtOH exposure?

Answer: As the reviewer suggested, we additionally measured kynurenine levels in the liver and sera of *Ahr* HKO mice. We found significantly increased hepatic kynurenine levels in *Ahr* HKO mice treated with ethanol compared to those with vehicle. However, serum kynurenine levels were comparable to each other. Although mechanistic studies are additionally required, we carefully raise the possibility that no changes in the levels of serum kynurenine under the alcohol condition in *Ahr* HKO animals may be due to multiple factors such as altered metabolite transport, transporter expression, and kynurenine utilization. The data was included in the revised manuscript (revised Supplementary Fig. 3d).

4. The effect of AhR on autophagy was an association, the authors did not provide sufficient data to show the functional relevance of autophagy in phenotypic exhibition.

Answer: The authors thank the reviewer for the valuable comment and agree with the reviewer’s point. To determine whether AhR attenuates ethanol-induced lipid accumulation via autophagy regulation, we co-treated *Ahr* HKO mouse primary hepatocytes with chloroquine (CQ; an autophagy inhibitor). Indeed, CQ co-treatment causes accumulation of lipid droplets compared to AhR deletion alone (revised Supplementary Fig. 6h). In addition, we already measured oxygen consumption rates (OCRs) in WT or *Ahr* HKO mouse primary hepatocytes treated with CQ, and confirmed that the OCRs increased by AhR loss was diminished by CQ to the degree of WT control (revised Fig. 6i). All of the results support the relevance of AhR-dependent autophagy to phenotypic changes. This was discussed in the revised manuscript. We hope that our efforts satisfy the reviewer’s concern.

5. AhR is differentially expressed in different liver cell types. It is necessary to provide data showing AhR expression in different liver cell types (hepatocytes, hepatic stellate cells, Kupffer cells, and sinusoidal endothelial cells) upon alcohol challenge. Does AhR in other liver cells play a role in ALD?

Answer: The authors appreciate the reviewer's valuable comment. In hepatic stellate cells, it has been shown that ethanol treatment activates AhR and down-regulates its level (Zhang, H. F. *et al.*, *Alcohol. Clin. Exp. Res.*, 2012). A very recent study also showed that down-regulation of AhR in hepatic stellate cells promoted alcohol-associated fibrosis (Schonfeld, M. *et al.*, *Hepatol. Commun.*, 2022). However, the roles of AhR in Kupffer cells and Liver sinusoidal endothelial cells were unclear. As the reviewer suggested, we treated hepatic stellate cells and Kupffer cells isolated from mice with ethanol and found decrease and increase of AhR in the former and in the latter cells, respectively. The data was added (revised Supplementary Fig. 1d) and stated in the revised manuscript.

6. The legends of Extended Data Figure 2 a-d are incorrect.

Answer: As the reviewer pointed out, the legends of revised Supplementary Fig. 2 were corrected.

7. It is necessary to provide mRNA or protein level of AhR in hepatocytes from HEP-AhR knockout mice vs. flox controls to verify the AhR knockout efficiency in hepatocytes.

Answer: As the reviewer pointed out, the AhR knockout efficiency in hepatocytes was confirmed by immunoblotting for AhR in the liver, hepatocytes, and hepatic stellate cells. The results were included in revised Supplementary Fig. 2c.

8. What is the effect of kynurenine in liver autophagy in mice fed on Lieber-DeCarli diet?

Answer: We further examined the effect of kynurenine on autophagy in hepatocytes exposed to alcohol with or without IDO/TDO2 inhibitors. Primary hepatocytes isolated from WT mice fed a Lieber-DeCarli diet for 5 weeks were treated with IDO/TDO2 inhibitors and continuously exposed to kynurenine (or vehicle). As expected, treatment with IDO/TDO2 inhibitors enhanced autophagy flux, which was diminished by kynurenine, confirmative of kynurenine effect on autophagy. The data was included in revised Supplementary Fig. 3g.

9. What is the effect of IDO/TDO2 knockdown/inhibition on hepatocytes isolated from Lieber-DeCarli diet-treated mice?

Answer: The authors thank the reviewer for the valuable comment. As the reviewer suggested, we additionally examined the effects of IDO/TDO2 inhibitors treatment on the identified targets in association with autophagy and confirmed the expected outcomes using hepatocytes isolated from Lieber-DeCarli diet-treated mice. The results show the causal relationship between the identified targets and kynurenine-mediated AhR activation. The data were included in revised Supplementary Fig. 5j.

10. Please explain why AHR mRNA expression was not changed between ALD and Normal liver (Figure 7a), but the protein expression was increased (Figure 7c).

Answer: The authors appreciate the reviewer's helpful comment. As the reviewer pointed out, the results between AhR mRNA and protein expression showed a discrepancy, unlike the case in mice. As we already discussed in the manuscript, the ALD mouse model may have a limitation to understand the pathogenesis of alcoholic hepatitis in clinical situations. AhR is also post-translationally regulated: ubiquitination and SUMOylation (Ma, Q. *et al.*, *J. Biol. Chem.*, 2000; Xing, X. *et al.*, *J. Cell. Physiol.*, 2012). Related to this, further studies remain to better understand the mechanism as to the post-translational control of AhR, which may explain why AhR mRNA levels were unchanged in ALD patients. This was stated in the revised manuscript.

Reviewer #2

Authors investigated the role and potential mechanisms of aryl hydrocarbon receptor (AhR) in regulating AMPK and autophagy in the early stage of alcoholic liver disease (ALD). They utilized a Lieber-DeCarli alcohol diet feeding mouse model that can mimic early stage of steatosis of ALD as well as in vitro cell culture (primary hepatocytes and AML12) and correlative studies of human ALD

liver samples. Unbiased transcriptomic and metabolomics studies were used. They identified increased levels of kynurenine, an endogenous AhR ligand in ALD mouse models and ethanol-treated hepatocytes. They further identified Ppp2r2d, a regulatory subunit of PP2A, as a transcriptional target of AhR, which was upregulated in alcohol fed mouse livers resulting in decreased phosphorylation of AMPK and possible decreased hepatic autophagy. Liver-specific AhR knockout mice fed with alcohol diet also had decreased hepatic steatosis and liver injury. The authors concluded that kynurenine-mediated AhR activation by alcohol may inhibit autophagy flux in hepatocytes through Ppp2r2d-mediated AMPK α dephosphorylation. Overall, the identification of AhR- Ppp2r2d-AMPK α axis in regulating autophagy in ALD is novel and interesting. Most data were of good quality and comprehensive/complementary to support the role of AhR in the early ALD. However, several limitations/weaknesses were noticed for this study as the mouse model only can reflect the early stage but not the more severe stage of ALD, and the later (such alcoholic hepatitis) is more important in clinic. Some of the data were also questionable and need to be further improved/clarified.

Specific comments:

1. Figure 1G, better quality IHC staining images should be provided for AHR (hardly to see any nuclear AHR staining). It seems that CYP1A1 staining showed a clear zonation pattern. Was CYP1A1 staining enriched in the central vein area? Ideally authors should show the same liver areas for the IHC staining of both AHR and CYP1A1 to show the correlation.

Answer: The authors express sincere thanks to the reviewer for the meticulous review. According to the comment, the histological images of AhR were switched with better high-resolution images (revised Fig. 1g). AhR signaling is associated with zoned expression patterns of CYP1A1 in liver lobules; if AhR is inhibited, the area of CYP1A1 expression around the central vein is small, whereas it extends larger into periportal regions when AhR is activated (Schulthess, P. *et al.*, *Nucleic Acids Res.*, 2015). As the reviewer suggested, we performed immunofluorescent staining to determine the colocalization of AhR with CYP1A1, and found that AhR-positive cells co-expressed CYP1A1. This was stated in the revised manuscript (revised Supplementary Fig. 1c).

2. Supplemental Figure 2, in addition to the genotyping PCR data, western blot analysis data for liver AHR should be provided (even if some truncated proteins may be present as authors claimed). Supplemental Figure 2E, the statement "Alcohol exposure inhibited the autophagy markers" was too vague. Authors should perform densitometry analysis and clearly stated the changes on LC3B-II and p62 after alcohol feeding.

Answer: As the reviewer pointed out, the *Ahr* knockout efficiency in hepatocytes was verified by immunoblotting for AhR in the liver, hepatocytes, and hepatic stellate cells. The data was included in revised Supplementary Fig. 2c. Also, as the reviewer suggested, we added densitometric values in the revised manuscript (revised Supplementary Fig. 2g), which clearly shows changes in LC3B-II and p62. Moreover, according to the reviewer's comment, the expression was changed from "Alcohol exposure inhibited the autophagy marker" to "alcohol treatment decreased LC3B II levels with increase of p62, confirming reduced autophagy flux, which is in line with previous reports".

3. Figure 2C, the quality of the EM image was poor and hardly to see any autophagy-related structures. Why the control *Ahr* HKO mice had a lot of LDs, which was not consistent with Figure 6b? Was the image representative as *Ahr* HKO mice. A better rigorous controlled autophagic flux assay was lacking for 2D, F, G, as without a lysosomal inhibitor these data were difficult to interpret. As it would be difficult to perform this in vivo (requires another feeding), authors should consider at least to repeat the autophagic flux assay in vitro primary hepatocytes and AHR knockdown AML12 cells with ethanol treatment and with/without chloroquine.

Answer: The authors thank the reviewer for the helpful suggestion. According to the reviewer, the TEM images were switched with better high-resolution images (revised Fig. 2c). We also confirmed that hepatic deletion of AhR diminished the ability of alcohol to increase hepatic and serum triglycerides contents in not only Lieber-DeCarli (revised Fig. 6b), but also the chronic plus binge NIAAA model (ALD mouse models) (revised Supplementary Fig. 7h). Moreover, using the TEM method, we observed that accumulation of lipid droplets was diminished in *Ahr* HKO mouse liver. Therefore, we

replaced the representative TEM image of *Ahr* HKO, the result of which matches with physiological phenotype changes. We further examined the effect of AhR under ethanol conditions on autophagy regulation, and added the data in the revised Supplementary Fig. 2k. As expected, accumulation of LC3-II by CQ was higher in AhR-deficient cells than in control cells upon ethanol treatment, confirmative of AhR effect on autophagy flux. We hope that our explanations, changes, and supplementary experiments would make you and other readers much easier to understand the manuscript.

4. The levels of p62 in *Ahr* KO hepatocytes and KD AML-12 cells was almost undetectable and very striking (Figure 2G and Figure 3I). As it is well known p62/SQSTM1 is also subjected to transcription regulation, it will be helpful to provide the mRNA data of p62/SQSTM1 given AHR is a transcription factor. On the same line, it will be interesting to see whether other autophagy-related genes would be changed at the transcription levels after manipulating AHR such as providing a heatmap for autophagy-related genes from the RNAseq analysis? Figure 3H, p62 levels should be provided to complete the experiments.

Answer: The authors thank the reviewer. The authors agree with the comment that AhR may affect p62 mRNA. In the supplementary experiment, we examined the possible transcriptional effect of AhR using primary hepatocytes; p62 mRNA levels were minimally changed by the loss of AhR (revised Supplementary Fig. 2i), supporting that p62 may not be the direct target of AhR. As the reviewer suggested, the autophagy-related genes were analyzed using the hepatic transcriptome data from WT and *Ahr* HKO. Gene Ontology analysis and heatmaps using upregulated 514 DEGs affected by AhR deletion systematically revealed that 33 genes related to autophagy may be under the control of AhR. In addition, we confirmed the absence of p62 in that gene cluster. The data was added in revised Supplementary Fig. 2f and discussed in the revised manuscript. Also, p62 levels were included in revised Fig. 3h.

5. Figure 6, the improved mitochondria biogenesis in *Ahr* HKO mice was interesting. It was known that TFEB regulated PGC1 α and biogenesis of mitochondria and lysosome, which is impaired by alcohol. This should be discussed or explored. Autophagy is a dynamic process and autophagic flux can be influenced by the early autophagosome synthesis or late-stage lysosomal functions/biogenesis, and it has been reported that alcohol consumption may affect the later state autophagy too (PMID:29782848). This should be discussed to clearly reflect the progress of this current study to the ALD field. AMPK regulates autophagy by inhibiting mTORC1 activity or ULK1 phosphorylation? These downstream events after alcohol should be explored to provide additional mechanistic insights.

Answer: As the reviewer pointed out, TFEB controls the metabolic response by promoting PGC1 α -mediated lipid metabolism (Settembre, C. *et al.*, *Nat. Cell Biol.*, 2013) and is transcriptionally regulated by PPAR α (Ghosh, A. *et al.*, *J. Biol. Chem.*, 2015). Moreover, AhR inhibits hepatic PPAR α (Shaban, Z. *et al.*, *Xenobiotica.*, 2005). Thus, it is speculated that the TFEB pathway might be a possible mechanism underlying the effect of AhR on mitochondria and lipid metabolism in alcohol conditions. As the reviewer mentioned, alcohol may affect the later stage of autophagy (Chao, X. *et al.*, *Gastroenterology*, 2018). However, the role of autophagy in acute and chronic ALD is still complex and contentious. In a mouse model with Lieber-DeCarli diet containing different concentrations of alcohol for 4 weeks, autophagy was facilitated by a lower dose of alcohol (29% ethanol-derived calories) but abrogated by a higher dose (36% ethanol-derived calories) (Lin, C. W. *et al.*, *J. Hepatol.*, 2013). Moreover, some studies present elevated levels of LC3 and p62 derived from the dysfunction of lysosomes (Thomes, P. G. *et al.*, *Alcohol. Clin. Exp. Res.*, 2015; Manley, S. *et al.*, *Redox. Biol.*, 2014), whereas others including ours report decreased levels of LC3 *in vivo* or *in vitro* (Noh, B. K. *et al.*, *Biochem. Biophys. Res. Commun.*, 2011; Lee, Y. J. *et al.*, *PLoS One*, 2019). This discrepancy could be explained by methodologic differences, such as the length of alcohol exposure, the concentration of alcohol, and frequency/routes of alcohol administration. Thus, the authors carefully surmised that AhR may affect the early stage of autophagy in our models. As the reviewer suggested, we measured the effect of AMPK on ULK1 regulation in hepatocytes treated with alcohol, and the data was added in the revised Supplementary Fig. 4e. ULK1 phosphorylation by AMPK is required for ULK1 activation in autophagy (Egan, D. F. *et al.*, *Science*, 2011; Hong-Brown, L. Q. *et al.*, *Alcohol. Clin. Exp. Res.*, 2017). Consistent with previous reports, our experiment using AMPK

activator/inhibitor showed that ULK1 phosphorylation is controlled by AMPK under alcohol conditions. The phosphorylation of ULK1 was also increased in *Ahr* HKO mouse primary hepatocytes (revised Supplementary Fig. 4f). So, it is assumed that the decrease in ULK1 activity by alcohol may reflect the inhibitory effect of AhR on AMPK. We think it may be worthwhile to investigate the reviewer suggestions (i.e., AhR-TFEB or AhR-AMPK-ULK1 axis). However, this requires substantial additional work which is beyond the current scope of the study. This was stated in the revised manuscript.

6. For the human ALD samples, more detail description on these ALD samples should be provided as ALD ranges from early steatosis to later stage fibrosis, inflammation and alcoholic hepatitis. This is particular important, as the "ALD mouse model" by Lieber-DeCarli diet feeding only represent/recapitulate the early steatosis phenotype of ALD (mild or no inflammation and fibrosis). This is a limitation of this study. As the human ALD samples seem to be from late stage ALD based on the sole H&E staining imaging data (Supplemental figure 7A). The impact of this manuscript may be improved if author can apply other relative more severe ALD mouse models such as the chronic plus binge NIAAA model.

Answer: The authors thank the reviewer for the valuable discussion and helpful comments. The authors agree with the referee's comment. As the reviewer suggested, detailed information on the human ALD samples was included in the revised Supplementary Table 1. As an additional effort, the statistical analysis of clinical parameters was performed according to the stages of steatosis or fibrosis (revised Supplementary Table 2, 3). Moreover, as expected, linear regression analysis showed a significant correlation between the fibrosis stage and all identified targets. The additional data was added in revised Supplementary Fig. 8c and discussed in the revised manuscript. According to the reviewer's comment, we additionally used the chronic plus binge NIAAA model. Similar to the Lieber-DeCarli model, chronic-binge alcohol treatment induced significantly higher levels of AhR, CYP1A1, IDO, and TDO2. We also confirmed increases in PPP2R2D level, hepatic steatosis, TGs, and liver injury and the effect of AhR depletion on p-AMPK and autophagy. Collectively, these results support the contention that AhR ablation protects hepatocytes during late-stage ALD. The results were included in revised Supplementary Fig. 7 and discussed.

7. Another limitation for this study was that authors should be aware that both AHR and AMPK have multiple targets/pathways rather than regulating autophagy, so other pathway may be involved for the ALD phenotypes of AHR KO mice that they observed in addition to autophagy. This should be discussed.

Answer: The authors appreciate the reviewer's helpful comment. The authors agree with the comment that there is a possibility of other pathways involved in ALD phenotypes via the AhR/AMPK axis. It is well recognized that AMPK promotes de novo mitochondrial biogenesis through PGC1 α -dependent transcription and suppresses lipid accumulation by ACC activity inhibition in an autophagy-independent manner (Egan, D. F. *et al.*, *Science*, 2011; Hong-Brown, L. Q. *et al.*, *Alcohol. Clin. Exp. Res.*, 2017). Also, past studies have shown that AhR plays a role in the formation and decomposition of lipids. (Lee, J. H. *et al.*, *Gastroenterology*, 2010; Yao, L. *et al.*, *Hepatology*, 2016). In our original manuscript, the RNA-seq data showed that various metabolic pathways as well as the AMPK signaling pathway were regulated by the loss of AhR. Given this, although our data uncovered that AhR inhibits AMPK signaling and subsequently interrupts autophagy flux, we cannot rule out that other pathways regulated by AhR/AMPK directly or indirectly account for some ALD improvement. This was discussed in the revised manuscript.

8. Authors should do a better job to summarize/discuss on recently findings on Autophagy and AHR in ALD (some key papers were even not mentioned; PMID: 3408211, 34454169; 29782848).

Answer: As the reviewer mentioned, Wrzosek, L. *et al.* described the role of AhR in intestinal microbiota biology (*Gut*, 2021), suggesting that increased tryptophan metabolite in the gut attenuated alcoholic liver injury. This study focused on the tryptophan metabolite, specifically, indole as an AhR ligand produced by the gut microbiome. However, in mammals, more than 90% of tryptophan is degraded through the kynurenine pathway and so the liver is responsible for 90% of this event (Savitz, J., *Mol. Psychiatry*, 2020; Badawy, A. A., *Int. J. Tryptophan Res.*, 2017), indicating that the kynurenine

pathway is more critical to tryptophan catabolism. Although Wrzosek, L. *et al.* demonstrated that there was no difference in the serum level of kynurenine among patients with and without alcoholic hepatitis, serum tryptophan levels were lower in alcoholic hepatitis patients, intimating that the kynurenine/tryptophan ratio, reflecting IDO/TDO2 activity, would be increased (Badawy, A. A. *et al.*, *Int. J. Tryptophan Res.*, 2019). These results support our hypothesis that alcohol induces the kynurenine pathway via IDO/TDO2 in the liver. Also, they used AhR whole-body knockout mice exposed to a lower concentration of alcohol for a shorter period of time. So, the discrepancy could be due to methodologic differences. Furthermore, the gut-liver axis plays a role in driving the disease (Szabo, G., *Gastroenterology*, 2015). As the gut microbiome is a source of endogenous AhR ligands, not to mention the gut-liver axis, the produced AhR ligands may reach the liver via portal blood, affecting liver biology (Cella, M. *et al.*, *Semin. Immunol.*, 2015). Qian, M. *et al.* also showed that indole-3-carbonol, a tryptophan metabolite, activates intestinal AhR to improve the function of the intestinal barrier (*Cell. Mol. Gastroenterol. Hepatol.*, 2022). The study, however, did not address the AhR signaling in hepatocytes. Additionally, the authors used a mild/acute alcohol model (10 days administration), as reflected by lower ALT activity (20-30 U/L) compared to >120 U/L in ours. So, it seems that they studied the early stage of ALD pathology. Together, we carefully raise the possibility that the intestinal AhR ligand plays a role in the liver in early ALD, but along with ALD progression, excessively induced kynurenine and AhR in the liver may obscure the intestinal effect. According to the reviewer's comment, the most recent findings were cited and discussed.

Reviewer #3 (Remarks to the Author):

Kim *et al.* have demonstrated the role of AhR in regulation of lipid metabolism in alcohol-fed mice. The metabotropic AhR overinduction by alcohol inhibits autophagy in hepatocytes through AMPK α , as mediated by Ppp2r2d gene transactivation, revealing a new AhR-dependent metabolism of phospho/sphingo-lipids. To draw a more solid conclusion, several concerns need to be addressed.

1. Metabolic profiling of serum, liver (liver biopsy tissue) and feces from clinical cohort are necessary to validate the increased levels of kynurenine in ALD patients. Previous studies have proved that high productions of bacterial tryptophan metabolites were associated with an improvement of hepatic injury, which are not consistent with this study (e.g., PMID: 33004548).

Answer: We thank the reviewer for this great suggestion and agree that including the data would enhance the clinical significance. Unfortunately, due to limitations in the supply of additional human samples (serum, liver biopsy, or feces) necessary for the detection of kynurenine, further assays were impossible. Instead, there are other reports showing that alcohol induces kynurenine levels in human serum and urine (Badawy, A. A. *et al.*, *Alcohol Alcohol.*, 2009; Buydens-Branchey, L. *et al.*, *Alcohol. Clin. Exp. Res.*, 1988; Badawy, A. A., *Adv. Exp. Med. Biol.*, 1999). These were cited in the revised manuscript. As the reviewer mentioned, Wrzosek, L. *et al.* described the role of AhR in intestinal microbiota biology (*Gut*, 2021), suggesting that increased tryptophan metabolite in the gut attenuated alcoholic liver injury. This study focused on the tryptophan metabolite, specifically, indole as an AhR ligand produced by the gut microbiome. However, in mammals, more than 90% of tryptophan is degraded through the kynurenine pathway. So the liver may be responsible for 90% of this event (Savitz, J., *Mol. Psychiatry*, 2020; Badawy, A. A., *Int. J. Tryptophan Res.*, 2017), indicating that the kynurenine pathway is more critical in tryptophan catabolism. Although Wrzosek, L. *et al.* demonstrated that there was no difference in the serum level of kynurenine between patients with and without alcoholic hepatitis, serum tryptophan levels were lower in alcoholic hepatitis patients, intimating that the kynurenine/tryptophan ratio, reflecting IDO/TDO2 activity, would be increased (Badawy, A. A. *et al.*, *Int. J. Tryptophan Res.*, 2019). These results support the hypothesis that alcohol induces the kynurenine pathway via IDO/TDO2 in the liver. Also, they used AhR whole-body knockout mice exposed to a lower concentration of alcohol in a shorter period of time. Therefore, the discrepancy could be explained by methodologic differences. The most recent findings were cited and discussed.

2. Previous studies established the role of gut microbiota-metabolites in activation of intestinal AHR for the alleviation of alcohol-induced liver injury (PMID: 34454169; 33004548), which was not in

accordance with the hepatic outcomes. Please explain the discrepancy between the effects of AHR on liver and intestine or whether there is any crosstalk between the hepatic and intestinal AHR signaling.

Answer: As mentioned above, there are some differences depending on animal models. Some patient data, however, consist with ours. Admittedly, the gut-liver axis plays a role in driving the disease (Szabo, G., *Gastroenterology*, 2015). As the gut microbiome is a source of endogenous AhR ligands, not to mention the gut-liver axis, the produced AhR ligands may reach the liver via portal blood, affecting liver biology (Cella, M. *et al.*, *Semin. Immunol.*, 2015). Qian, M. *et al.* also showed that indole-3-carbinol, a tryptophan metabolite, activates intestinal AhR to improve the function of the intestinal barrier (*Cell. Mol. Gastroenterol. Hepatol.*, 2022). The study, however, did not address the AhR signaling in hepatocytes. Additionally, the authors used a mild/acute alcohol model (10 days administration), as reflected by lower ALT activity (20-30 U/L), compared to ours (>120 U/L). So, it seems that they examined the early stage of ALD. Together, we carefully raise the possibility that the intestinal AhR ligand plays a role in the liver in early ALD, but as ALD proceeds, excessively induced kynurenine and AhR in the liver obscure the intestinal effect. This was briefly discussed.

3. It is observed that hepatic expression of IDO was elevated in mice intervened with alcohol, thus, please provide some mechanistic links between the alcohol and IDO or TDO2.

Answer: The authors appreciate the reviewer's comment. Previous studies have shown that IDO was induced by IFN- γ in many types of cells including hepatocytes (Taylor, M. W. *et al.*, *FASEB J.*, 1991). Also, the link between alcohol and IFN- γ is well established (Frank, K. *et al.*, *PLoS One*, 2020; Pang, M. *et al.*, *BMC Immunol.*, 2011). In other studies, ethanol administration elevated plasma glucocorticoid concentration and promotes the activity of TDO2 in the liver (Rose, A. K. *et al.*, *Alcohol. Clin. Exp. Res.*, 2010; Kiseleva, I. P. *et al.*, *Farmakol. Toksikol.*, 1981). Hence, IDO and TDO2 levels seem to be controlled through different mechanisms. This was briefly stated in the revised manuscript.

4. As shown in Figure 3a, the "tyrosine metabolism" was significantly upregulated in ethanol-treated group compared with control, I just wonder whether "tyrosine metabolism"-related metabolites changed accordingly?

Answer: The authors thank the reviewer for the meticulous reading of our MS. We additionally performed Metabolite Set Enrichment Analysis (MSEA) using the serum and tissue metabolomics data and found that the metabolic pathways associated with tyrosine were also recognized among the top 20 most enriched pathways. The results were included in revised Supplementary Fig. 3e.

5. Please provide the immunoblotting results of p62 in Figure 3h.

Answer: As the reviewer suggested, p62 blottings were added in revised Fig. 3h.

6. Please provide the liver weights of WT and Ahr HKO groups other than the liver/body weight ratio.

Answer: As the reviewer recommended, the liver weights of WT and *Ahr* HKO groups were added in revised Supplementary Fig. 6e.

7. The tissue microarray putting all the human tissue samples into one array is recommended to see the immunohistochemical differences of the molecular targets between the two groups more clearly.

Answer: The authors thank the reviewer for this suggestion and agree that the suggested method will make our data more intuitive to understand. As limitations arise when large populations of required tissue samples are not available, tissue microarray is mainly applied for cancer research for which patient samples are relatively easy to obtain. Due to the limited supply of human samples in our research, we calculated H-scores using IHC images. We hope that our additional efforts would make you and other readers much easier to understand our results. The results were included in revised Fig. 7d.

8. The length of scale bars throughout the manuscript are not consistent although they all represent 100 μm .

Answer: As the reviewer pointed out, we adjusted the scale bars to the same length all over the images related to the mouse samples.

9. Seven mice in control group and 9 in alcohol-treated were included for the metabolomics analysis, why only three in each group were chosen for the RNA-seq analysis.

Answer: As the cost is too high, RNA-seq was done using three samples per group which is the minimal required number for most statistical tests. Admittedly, samples were randomly selected. This was added in the Methods section.

10. There are mistakes in the figure legends in Extended Data Figure 2, 4.

Answer: The legends of revised Supplementary Fig. 2 and 4 were edited.

I appreciate your kind consideration for disseminating the present findings in *Nature Communications*.

Sincerely Yours,

Sang Geon Kim, Ph.D., Professor

College of Pharmacy

Dongguk University-Seoul, Goyang 10326

Republic of Korea;

Tel: +8231-961-5218; Fax: +8231-961-5206; E-mail: sgkim@dongguk.edu

References for the answers

1. Heo, M. J. et al. Alcohol dysregulates miR-148a in hepatocytes through FoxO1, facilitating pyroptosis via TXNIP overexpression. *Gut* **68**, 708-720 (2019).
2. Ding, W. X. et al. Autophagy reduces acute ethanol-induced hepatotoxicity and steatosis in mice. *Gastroenterology* **139**, 1740-1752 (2010).
3. Nourissat, P. et al. Ethanol induces oxidative stress in primary rat hepatocytes through the early involvement of lipid raft clustering. *Hepatology* **47**, 59-70 (2008).
4. Schulze, R. J. et al. Ethanol exposure inhibits hepatocyte lipophagy by inactivating the small guanosine triphosphatase Rab7. *Hepatol. Commun.* **1**, 140-152 (2017).
5. Lee, Y. J. et al. Cilostazol protects hepatocytes against alcohol-induced apoptosis via activation of AMPK pathway. *PLoS One* **14**, e0211415 (2019).
6. Zhang, H. F. et al. Regulation of the activity and expression of aryl hydrocarbon receptor by ethanol in mouse hepatic stellate cells. *Alcohol. Clin. Exp. Res.* **36**, 1873-1881 (2012).
7. Schonfeld, M., Averilla, J., Gunewardena, S., Weinman, S. A. & Tikhanovich, I. Alcohol-associated fibrosis in females is mediated by female-specific activation of lysine demethylases KDM5B and KDM5C. *Hepatol. Commun.* <https://doi.org/10.1002/hep4.1967> (2022).
8. Ma, Q. & Baldwin, K. T. 2,3,7,8-tetrachlorodibenzo-p-dioxin-induced degradation of aryl hydrocarbon receptor (AhR) by the ubiquitin-proteasome pathway. Role of the transcription activator and DNA binding of AhR. *J. Biol. Chem.* **275**, 8432-8438 (2000).
9. Xing, X. et al. SUMOylation of AhR modulates its activity and stability through inhibiting its ubiquitination. *J. Cell. Physiol.* **227**, 3812-3819 (2012).
10. Schulthess, P. et al. Signal integration by the CYP1A1 promoter--a quantitative study. *Nucleic Acids Res.* **43**, 5318-5330 (2015).
11. Settembre, C. et al. TFEB controls cellular lipid metabolism through a starvation-induced autoregulatory loop. *Nat. Cell Biol.* **15**, 647-658 (2013).
12. Ghosh, A. et al. Activation of peroxisome proliferator-activated receptor alpha induces lysosomal biogenesis in brain cells: implications for lysosomal storage disorders. *J. Biol. Chem.* **290**, 10309-10324 (2015).
13. Shaban, Z. et al. AhR and PPARalpha: antagonistic effects on CYP2B and CYP3A, and additive inhibitory effects on CYP2C11. *Xenobiotica.* **35**, 51-68 (2005).

14. Chao, X. et al. Impaired TFEB-mediated lysosome biogenesis and autophagy promote chronic ethanol-induced liver injury and steatosis in mice. *Gastroenterology* **155**, 865-879.e12 (2018).
15. Lin, C. W. et al. Pharmacological promotion of autophagy alleviates steatosis and injury in alcoholic and non-alcoholic fatty liver conditions in mice. *J. Hepatol.* **58**, 993-999 (2013).
16. Thomes, P. G., Trambly, C. S., Fox, H. S., Tuma, D. J. & Donohue, T. M. Jr. Acute and chronic ethanol administration differentially modulate hepatic autophagy and transcription factor EB. *Alcohol. Clin. Exp. Res.* **39**, 2354-2363 (2015).
17. Manley, S. et al. Farnesoid X receptor regulates forkhead Box O3a activation in ethanol-induced autophagy and hepatotoxicity. *Redox. Biol.* **2**, 991-1002 (2014).
18. Noh, B. K. et al. Restoration of autophagy by puerarin in ethanol-treated hepatocytes via the activation of AMP-activated protein kinase. *Biochem. Biophys. Res. Commun.* **414**, 361-366 (2011).
19. Egan, D.F. et al. Phosphorylation of ULK1 (hATG1) by AMP-activated protein kinase connects energy sensing to mitophagy. *Science* **331**, 456-461 (2011).
20. Hong-Brown, L. Q., Brown, C. R., Navaratnarajah, M. & Lang, C. H. FoxO1-AMPK-ULK1 regulates ethanol-induced autophagy in muscle by enhanced ATG14 association with the BECN1-PIK3C3 complex. *Alcohol. Clin. Exp. Res.* **41**, 895-910 (2017).
21. Lee, J. H. et al. A novel role for the dioxin receptor in fatty acid metabolism and hepatic steatosis. *Gastroenterology* **139**, 653-663 (2010).
22. Yao, L. et al. Hyperhomocysteinemia activates the aryl hydrocarbon receptor/CD36 pathway to promote hepatic steatosis in mice. *Hepatology* **64**, 92-105 (2016).
23. Wrzosek, L. et al. Microbiota tryptophan metabolism induces aryl hydrocarbon receptor activation and improves alcohol-induced liver injury. *Gut* **70**, 1299-1308 (2021).
24. Savitz, J. The kynurenine pathway: a finger in every pie. *Mol. Psychiatry* **25**, 131-147 (2020).
25. Badawy, A. A. Kynurenine pathway of tryptophan metabolism: regulatory and functional aspects. *Int. J. Tryptophan Res.* **10**, 1178646917691938 (2017).
26. Badawy, A. A. & Guillemin, G. The plasma [kynurenine]/[tryptophan] ratio and indoleamine 2,3-dioxygenase: time for appraisal. *Int. J. Tryptophan Res.* **12**, 1178646919868978 (2019).
27. Szabo, G. Gut-liver axis in alcoholic liver disease. *Gastroenterology* **148**, 30-36 (2015).
28. Cella, M. & Colonna, M. Aryl hydrocarbon receptor: Linking environment to immunity. *Semin. Immunol.* **27**, 310-314 (2015).
29. Qian, M. et al. Aryl hydrocarbon receptor deficiency in intestinal epithelial cells aggravates alcohol-related liver disease. *Cell. Mol. Gastroenterol. Hepatol.* **13**, 233-256 (2022).
30. Badawy, A. A., Dougherty, D. M., Marsh-Richard, D. M. & Steptoe, A. Activation of liver tryptophan pyrrolase mediates the decrease in tryptophan availability to the brain after acute alcohol consumption by normal subjects. *Alcohol Alcohol.* **44**, 267-271 (2009).
31. Buydens-Branchey, L. et al. Increase in tryptophan oxygenase activity in alcoholic patients. *Alcohol. Clin. Exp. Res.* **12**, 163-167 (1988).
32. Badawy, A. A. Tryptophan metabolism in alcoholism. *Adv. Exp. Med. Biol.* **467**, 265-274 (1999).
33. Taylor, M. W. & Feng, G. S. Relationship between interferon-gamma, indoleamine 2,3-dioxygenase, and tryptophan catabolism. *FASEB J.* **5**, 2516-2522 (1991).
34. Frank, K. et al. Alcohol dependence promotes systemic IFN- γ and IL-17 responses in mice. *PLoS One* **15**, e0239246 (2020).
35. Pang, M., Bala, S., Kodys, K., Catalano, D. & Szabo, G. Inhibition of TLR8- and TLR4-induced Type I IFN induction by alcohol is different from its effects on inflammatory cytokine production in monocytes. *BMC Immunol.* **12**, 55 (2011).
36. Rose, A. K., Shaw, S. G., Prendergast, M. A. & Little, H. J. The importance of glucocorticoids in alcohol dependence and neurotoxicity. *Alcohol. Clin. Exp. Res.* **34**, 2011-2018 (2010).
37. Kiseleva, I. P., Lapin, I. P., Prakh'e, I. B. & Samsonova, M. L. Changes in hepatic tryptophan pyrrolase activity and plasma 11-hydroxycorticosteroid levels in rats after single and protracted alcohol administration and withdrawal. *Farmakol. Toksikol.* **44**, 319-322 (1981).

REVIEWERS' COMMENTS

Reviewer #1 (Remarks to the Author):

The authors have addressed many of my comments and the manuscript has improved from its previous version.

Among limitations, the use of the cell culture systems to study ALD has its limitation. Hepatoma cell lines are not relevant, while primary cells are marginally acceptable. Even use primary hepatocytes, the cell death is very mild even at high concentrations of EtOH. The readout of inflammation, which is a hallmark of ALD, is largely absent in the hepatocyte culture system. These limitations should be clearly discussed and emphasized.

Reviewer #3 (Remarks to the Author):

The authors have addressed all my comments with the original manuscript. The paper has been significantly improved after revising. These answers have been added in the discussion section. Although the metabolic profiling of clinical samples is not performed, it is acceptable due to sample accessibility. Moreover, the flaws in the results and figures have been revised. I recommend this paper for publication.

Response to Referees Letter

REVIEWERS' COMMENTS

Reviewer #1 (Remarks to the Author):

The authors have addressed many of my comments and the manuscript has improved from its previous version. Among limitations, the use of the cell culture systems to study ALD has its limitation. Hepatoma cell lines are not relevant, while primary cells are marginally acceptable. Even use primary hepatocytes, the cell death is very mild even at high concentrations of EtOH. The readout of inflammation, which is a hallmark of ALD, is largely absent in the hepatocyte culture system. These limitations should be clearly discussed and emphasized.

Answer: The authors thank the reviewer for the valuable discussion and helpful comments again. The authors agree with the referee's comment. As the reviewer suggested, we described this point in the latest version of manuscript as follows:

In the Discussion section, "In addition, our data using primary hepatocyte culture system supports ethanol-induced hepatic injury; Nonetheless, the outcomes have limitations in understanding ALD-associated cell death and inflammation because of its mild effect."

Reviewer #3 (Remarks to the Author):

The authors have addressed all my comments with the original manuscript. The paper has been significantly improved after revising. These answers have been added in the discussion section. Although the metabolic profiling of clinical samples is not performed, it is acceptable due to sample accessibility. Moreover, the flaws in the results and figures have been revised. I recommend this paper for publication.